# Self-association enhances early attentional selection through automatic prioritization of socially salient signals

**Meike Scheller[1,2]\*, Jan Tünnermann[3†], Katja Fredriksson[1], Huilin Fang[1], Jie Sui[1]\***

[1]University of Aberdeen, Aberdeen, United Kingdom; [2]Durham University, Durham, United Kingdom; [3]Philipp University of Marburg, Marburg, Germany

## eLife Assessment

This study presents a **valuable** finding on the mechanism of self-prioritization by revealing the influence of self-associations on early attentional selection. The evidence supporting the claims of the authors is **solid**, although inclusion of a discussion about the generalization and limitation would have strengthened the study. The work will be of interest to researchers in psychology, cognitive science, and neuroscience.

**\*For correspondence:**
meike.scheller@durham.ac.uk
(MS);
jie.sui@abdn.ac.uk (JS)

**Present address:** [†]Charlotte Fresenius University of Psychology, Cologne, Germany

**Competing interest:** The authors declare that no competing interests exist.

**Abstract** Efficiently processing self-related information is critical for cognition, yet the earliest mechanisms enabling this self-prioritization in humans remain unclear. By combining a temporal order judgement task with computational modeling based on the Theory of Visual Attention (TVA), we show how mere, arbitrary associations with the self can fundamentally alter attentional selection of sensory information into aware short-term memory, by enhancing the attentional weights and processing capacity devoted to encoding socially loaded information. This self-prioritization in attentional selection occurs automatically at early perceptual stages but reduces when active social decoding is required. Importantly, the processing benefits obtained from attentional selection via self-relatedness and via physical salience were additive, suggesting that social and perceptual salience captured attention via separate mechanisms. Furthermore, intra-individual correlations revealed an 'obligatory' self-prioritization effect, whereby self-relatedness overpowered the contribution of perceptual salience in guiding attentional selection. Together, our findings provide evidence for the influence of self-relatedness during earlier, automatic stages of attentional selection at the gateway to perception, distinct from later post-attentive processing stages.

## Introduction

The ability to prioritize self-related information is crucial for adaptive cognition and behavior. It enables us to efficiently process cues pertaining to our own safety, goals, and well-being in complex, social environments (*Conway, 2005*; *Enock et al., 2018*; *Humphreys and Sui, 2015*; *Moray, 1959*; *Rogers et al., 1977*). Decades of research have revealed that this self-relatedness boosts information processing not only for long-term established self-associated information (e.g. own names, owned objects), but also for completely arbitrary information newly associated with the self (*Cunningham et al., 2008*; *Golubickis et al., 2018*; *Moray, 1959*; *Scheller and Sui, 2022b*; *Sui et al., 2012*). However, the mechanisms underlying such self-prioritization remain unclear. While higher-level contributions from memory, decision making, and motor planning to the emergence of these self-prioritization effects (SPEs) are well established (*Caughey et al., 2021*; *Constable et al., 2011*; *Desebrock et al., 2018*; *Falbén et al., 2020a*; *Falbén et al., 2020b*; *Scheller and Sui, 2022b*; *Yin et al.,*

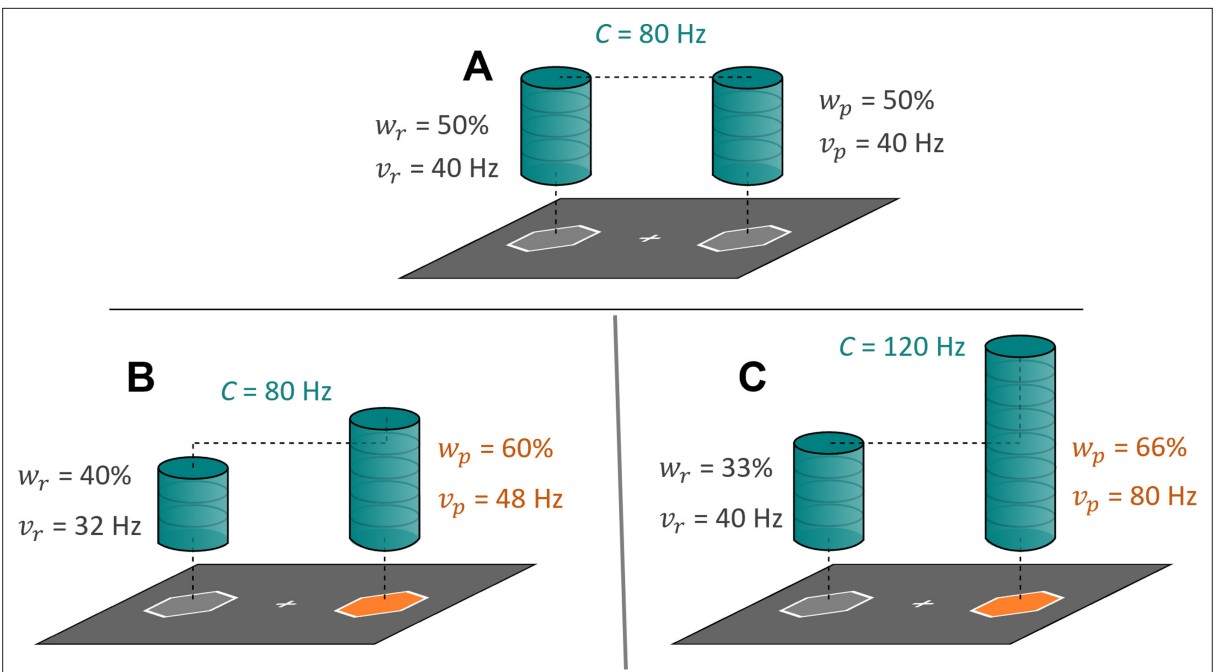

**Figure 1.** Mechanisms of attentional selection. Attentional selection can occur via different mechanisms: changes in the allocation or the availability and allocation of attentional resources. Panel **A** shows an example stimulus display with two identical perceptual objects (gray hexagons) on a dark background, and the processing resources ($C$, blue columns) that are distributed equally across these stimuli. In this case, the attentional weights for the reference ($w_r$; left stimulus) and probe ($w_p$; right stimulus) are identical. The processing rates for the reference ($v_r$) and probe ($v_p$) stimuli are given as the processing resources that are allocated to each of the two stimuli, i.e. $w_{p,r} * C$. Panels **B** and **C** show example stimulus displays with two perceptual objects, where one has a higher perceptual salience (luminance and color contrast). Panel **B** indicates a mechanism whereby the same processing resources are distributed differentially across the stimulus display, with more resources being given to the more salient stimulus. This is reflected in a change of relative weight, with constant processing capacity. Panel **C** indicates a mechanism whereby the amount of processing resources increases, along with a differential distribution of these resources. To arbitrate between these two mechanisms, we employed model comparisons to assess whether changes in relative attentional weight or changes in absolute processing rates (capacity and weights) better explained experimental data.

*2019*), the role of earlier, perceptual processing stages is still debated (*Macrae et al., 2017*; *Macrae et al., 2018*; *Reuther and Chakravarthi, 2017*; *Scheller and Sui, 2022a*; *Scheller and Sui, 2022b*; *Stein et al., 2016*; *Sui et al., 2012*). Some evidence suggests that self-relatedness can alter perceptual representations specifically through the integration of the sensory input with prior expectations (*Scheller et al., 2024*; *Scheller and Sui, 2022a*) or through attentional modulations (*Humphreys and Sui, 2016*; *Macrae et al., 2018*). Nevertheless, its specific effects on attentional selection during perception remain unanswered, particularly since socially associated stimuli often do not contain inherent sensory salience that automatically captures attention in a purely bottom-up fashion. The putative role of social salience in driving self-prioritization (*Humphreys and Sui, 2015*; *Liu and Sui, 2016*; *Moradi et al., 2020*; *Siebold et al., 2015*; *Sui et al., 2012*; *Sui et al., 2015*) begs the question whether the active decoding of higher-order social identities is strictly required to drive attentional selection, or whether more 'automatic' effects may arise from modulations of early perceptual attentional deployment.

The first goal of the present study was to outline the mechanisms by which self-relatedness influences attentional selection of visual information. To address this, we assessed whether and how social association of arbitrary, sensory information alters attentional selection, leading to prior entry (*Schneider and Bavelier, 2003*; *Spence and Parise, 2010*; *Titchener, 1908*; *Weiß et al., 2013*). Prior entry refers to the phenomenon that attention can boost processing of a stimulus so that it can sometimes be perceived as appearing earlier than another one, even if the other one was presented earlier. The magnitude of this catching up and overtaking in processing can be used to index the differential assignment of attentional resources to the stimuli. As a model of attention-biased stimulus encoding, the Theory of Visual Attention (TVA; *Bundesen, 1990*) formally describes mechanisms underpinning prior entry (; *Tünnermann et al., 2017*; *Figure 1*).

Prior entry could arise in different ways: (1) There may be a mere change in relative attentional weights (TVA's parameter *w*), where the attended stimulus receives resources at the expense of the other stimuli, with the overall employed processing resources (TVA's capacity parameter *C*) remaining constant. (2) According to TVA, objects in the visual field progress toward encoding at different processing rates (TVA's *v* parameters). Therefore, alternatively, there may be an absolute boost of processing resources to the advantage of the attended stimulus (or an absolute decrease of processing resources towards the unattended stimuli, see *Tünnermann et al., 2015*). In combination, these lead to a higher absolute processing rate v of the attended compared to the unattended stimulus and consequently to prior entry. By estimating TVA parameters with a hierarchical Bayesian model (*Tünnermann et al., 2017*), the present study quantified whether SPEs can be explained by changes in relative attentional weights, or absolute changes in processing rates of the self-related (salient) versus other-related (non-salient) stimuli. These mechanisms have previously been used to explain effects of low-level perceptual salience (*Krüger et al., 2016*; *Krüger et al., 2017*; *Krüger and Scharlau, 2021a*).

The second aim of the present study was to establish at what representational level (perceptual, social) visual attention spreads across stimuli with varying different degrees of social salience. Prior work suggests that self-relatedness of owned objects (*Constable et al., 2019*; *Truong et al., 2017*) and physical features (self-face; *Jublie and Kumar, 2021*) can lead to prior entry. While the mechanistic underpinnings of this attentional capture by self-relatedness have not been directly investigated, the authors manipulated the decisional dimension to be orthogonal to the social association. Findings from one study (*Constable et al., 2019*) suggested that self-prioritization only emerged when the self was a relevant decisional category. Crucially, though, their manipulation was also orthogonal to the perceptual feature of interest (the color of the object, rather than the location, which was random). Expecting self-prioritization under such conditions, however, would assume not just a perceptual modulation, but rather a modulation of a completely unrelated feature. This, in turn, assumes an automatic fusion of social representations across all its instantaneous, unrelated perceptual features. However, it is well-established that information that is transferred via different cues does not instantly underlie fusion of all its features (*Enock et al., 2018*). Hence, it remains unclear whether self-relatedness leads to prior entry at the level of the perceptual feature or requires decoding of the associated identity in a social feature dimension.

Lastly, the third and final aim of the present study was to qualitatively compare the effects of and quantitatively probe the interactions of perceptual and social salience. Using a temporal order judgement (TOJ) task with simple, colored shape stimuli arbitrarily associated with social identities, the present study can, for the first time, directly compare social and perceptual salience on the same mechanistic metrics. That is, qualitatively, we can assess similarities/differences in social and perceptual changes while, quantitatively, studying the interactions between social and perceptual salience. Notably, if social and perceptual salience operate completely independently, such as at different processing levels, one would expect no interaction between social and perceptual salience. On the other hand, if social associations systematically alter the effects of perceptual salience, this would be strongly suggestive of self-relatedness directly affecting bottom-up perceptual processing.

In summary, to characterize self-prioritization in early attentional selection, the study addresses three main research questions across two experiments. The resulting hypotheses and hypothesis-relevant details will be outlined more explicitly below:

**(Q1) Does mere self-association bias early attentional selection, and what are the underlying mechanisms? (Experiments 1+2)**

(H1) Based on previous findings of self-related prior entry findings (*Constable et al., 2019*; *Jublie and Kumar, 2021*; *Truong et al., 2017*), we hypothesized that participants would show an attentional selection bias towards self-associated, relative to other-associated information. Mechanistically, this could arise from an enhancement of relative attentional weights towards self-associated information (TVA parameter $w_p$) or an increase in processing rates for self-associated over other-associated information. To arbitrate between these possible mechanisms, we compared hierarchical Bayesian models estimating: attentional weights for each condition with a single processing capacity parameter across conditions (indicative of changes in relative attentional weights), and attentional weights and processing capacity independently for each condition (indicative of changes in absolute processing rates).

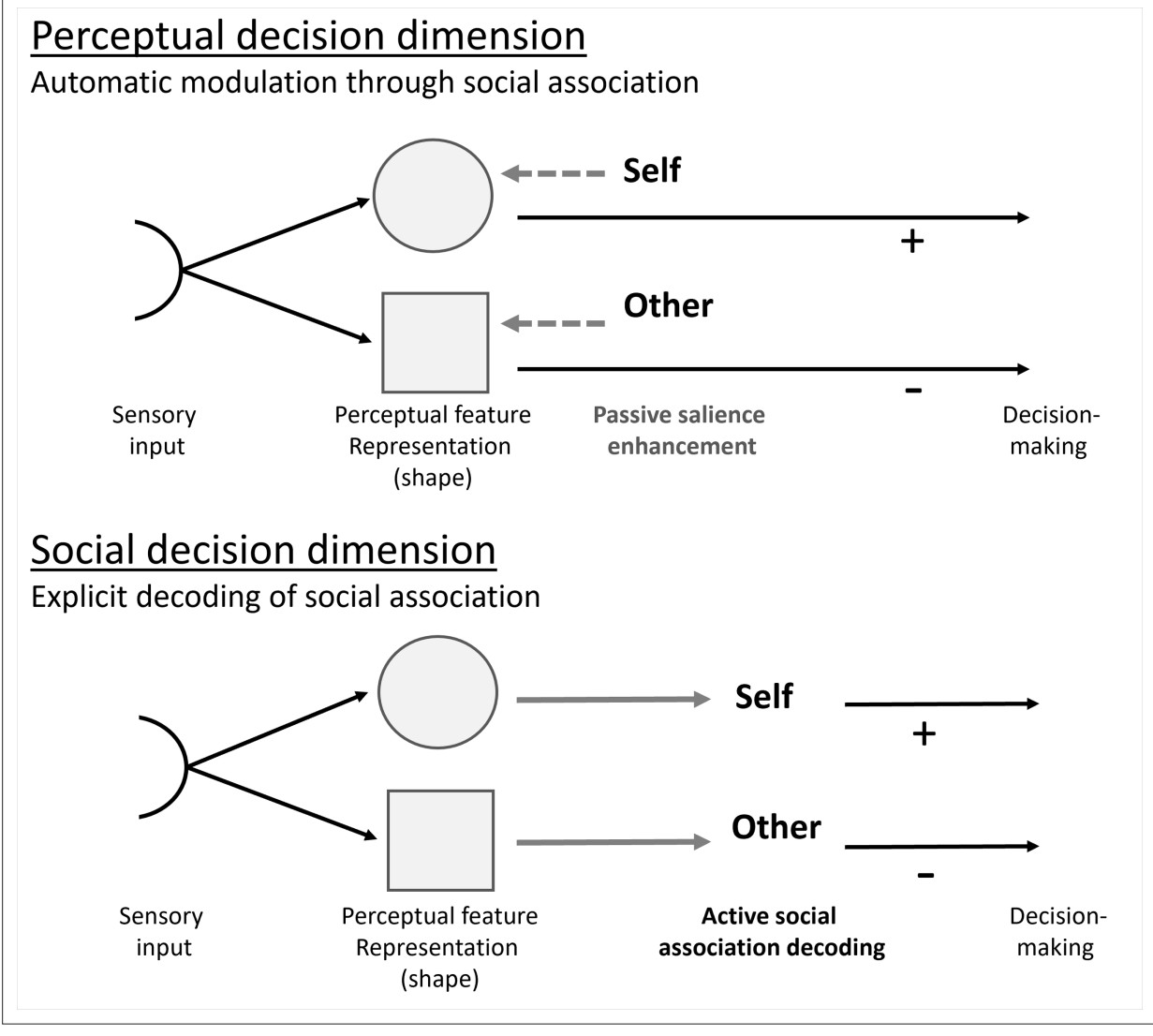

**Figure 2.** Decision dimensions. Dissociating the processing stages at which social association may affect early attentional selection via different decisional dimensions. Automatic effects of self-association would assume that active decoding of the associated social identity is not necessary. In this case, the social identity associated with a specific perceptual feature renders this feature more salient, without having to be consciously recalled. On the other hand, some studies suggested that the self needs to be a decisional criterion. In this case, self-prioritization effects in attentional selection would require active decoding of the social associations. Altering the decisional dimension (asking which shape vs whose shape), without shifting attention from the crucial perceptual feature (shape), allows disentangling these processes. Note that the directionality of the sensory and social information does not make assumptions about the temporal dynamics of the underlying process.

**(Q2) Does self-relatedness bias attentional selection automatically or does it require explicit social decoding? (Experiment 1)**

(H2) Based on previous findings (*Constable et al., 2019*; *Jublie and Kumar, 2021*; *Truong et al., 2017*), we hypothesized that participants would show a bias toward self-associated information when the decisional dimension requires the explicit decoding of the social identity (*Figure 2*, lower panel). If self-relatedness also biases attentional selection automatically at the perceptual feature level, self-related information should show higher relative attentional weights/absolute processing rates even when the decisional dimension does not require the explicit decoding of social identities, but merely of their associated perceptual feature (shapes; *Figure 2*, upper panel). As decisional criteria influence the expression of SPEs (e.g. *Caughey et al., 2021*; *Falbén et al., 2020b*; *Scheller and Sui, 2022b*), this automatic effect at perceptual levels was predicted to be smaller.

**(Q3) Does self-association affect attentional selection in a similar way to perceptual salience, and how do social and perceptual salience interact? (Experiment 2)**

(H3) If social and perceptual salience bias attentional selection in similar ways, both manipulations were hypothesized to result in enhancements of relative attentional weights or similar enhancements in absolute processing rates towards the more salient stimulus (self). Overall, we expected effects of social salience to be smaller than those of perceptual salience (*Liu and Sui, 2016*; *Mevorach et al., 2010*; *Sui et al., 2015*).

Social and perceptual salience may interact or be processed independently. If they operate completely independently, the combination of their attentional processing rate changes would be additive. Sub- or supra-additive effects of social and perceptual salience would suggest interacting processes during attentional selection. The difference in degree of additivity for perceptual salience with either self- or other-related information (interference of social and perceptual salience) would suggest that self-relatedness affects information processing via distinct attentional streams from information linked to other social identities.

Using a TOJ task with stimuli in which shape features have been arbitrarily associated with the self and a stranger identity (*Figure 3*), we measured whether mere social associations lead to a differential allocation of attention across the visual field, or an increase/decrease in processing rates for self- and other-associated stimuli. Perceptual salience effects were quantified within the same task and stimuli, by altering local color features. Baseline TOJ measures were conducted for each participant, thereby allowing control for individual, pre-existing biases towards specific perceptual features. Hence, reported difference scores between baseline and social association conditions (i.e. in $w_{peffect}$, $\Delta v_p$, or $\Delta v_r$) are directly indicative of processing changes resulting from social/perceptual salience.

Furthermore, to establish the extent to which early attentional selection contributes to frequently observed SPEs using shape-label matching tasks (*Sui et al., 2012*), we determined SPEs via this well-established paradigm. Employing the shape-label matching paradigm allowed us to form and practice associations between shapes and identities, while at the same time providing a (crude) independent measure of individual SPEs in matching. Within this task, self-relatedness biases build up over several processing stages including perception, attention, memory, and decision-making, leading to a behaviorally meaningful response-facilitation towards stimuli associated with the self. Group-level SPE measures were used to assess whether the included stimuli elicit self-prioritization benefits in the present sample, while, at the individual level, SPE measures were regressed over TVA parameters that indicated self-biases. The latter allowed us to determine whether the effects that attentional selection elicits in the perceptual or social representations contribute substantially to the SPE observed in the shape-label matching performance.

## Material availability statement

Research questions, analyses, models, and additional details have been preregistered on the Open Science Framework: https://osf.io/ehu75. The analysis notebooks and data are also available via the OSF project repository: https://osf.io/a62df.

## Results

### Does mere self-association lead to a bias in early attentional selection, and what are the underlying mechanisms?

To assess whether and how self-association leads to biases in early attentional selection, we estimated TVA-based parameters within a hierarchical Bayesian estimation procedure. Parameters were estimated by embedding the TVA-based TOJ equation (*Equation 1*) for the probability of reporting the probe stimulus as appearing first in hierarchical Bayesian models (*Tünnermann et al., 2015*). The model structure (*Figure 4*) shows that parameters were estimated, for each participant and each condition: the neutral baseline condition (N) and the social salience condition (S) with the social decision dimension. Within both conditions, the self-associated shape (salient) was defined as the probe, and the other-associated shape (non-salient) was defined as the reference. Crucially, the neutral baseline condition was used as an individual-specific correction, allowing us to measure the effects that were induced by social salience alone. Hence, any individual-specific biases or processing differences

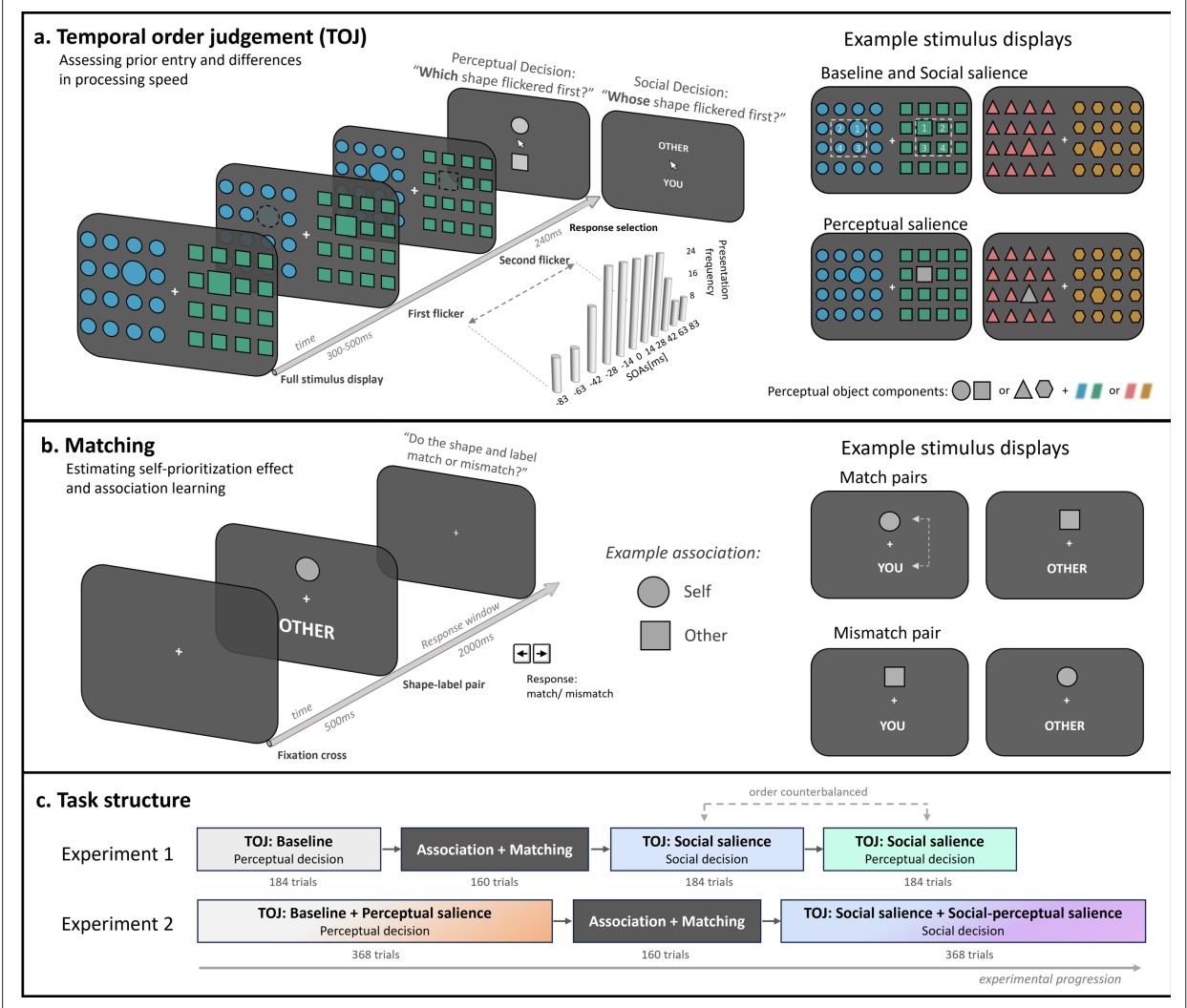

**Figure 3.** Task design. (**a**) Temporal order judgement task (TOJ) design. Following an initial presentation of the complete stimulus array, target shapes, which were relatively larger in size compared to background shapes, flickered with a variable stimulus onset asynchrony that was systematically varied between -/+ 83 ms with a higher presentation frequency at small SOAs. After the stimulus presentation, participants had to indicate which of the two shapes flickered first by selecting the correct shape (baseline conditions, perceptual salience conditions, social salience condition with perceptual decision boundary), or the identity label of the shape-associated social identity (social salience condition with social decision boundary). Stimulus displays consisted of two types of colored shapes (perceptual objects), distributed across two hemifields in an 8 x 8 grid. Targets would appear on each side at either of the four central locations. Lateralization of the specific perceptual objects was randomized across trials. (**b**) Perceptual matching task design. Participants associated one of the two shapes with themselves, and one with another, anonymous participant. Associations between social identities and perceptual objects were counterbalanced across participants. Pairs of shapes and social identity labels were presented on screen. These could either be congruent (matching) or incongruent (mismatching). Participants had to respond whether the pair matched in the learned association or mismatched. Location of the shapes and labels (above, below fixation) was counterbalanced across the task. (**c**) Task structures for Experiments 1 and 2. Both experiments began with a TOJ baseline task. Experiment 1 utilized non-salient targets exclusively, while Experiment 2 included both perceptually salient and non-salient targets. These were presented in randomly intermixed order. Next, targets were associated with social identities through a matching task. Following this association learning phase, which establishes social salience in the shapes, participants completed the same TOJ task again. In Experiment 1, they completed one block using a social decision dimension and one block using a perceptual decision dimension. The order of these blocks was counterbalanced across participants to reduce the influence of order effects in the results. In Experiment 2, perceptually salient and non-salient stimuli were presented in an intermixed fashion, and participants responded within the social decision dimension. Each task block was preceded by 8 (matching) to 14 (TOJ) practice trials.

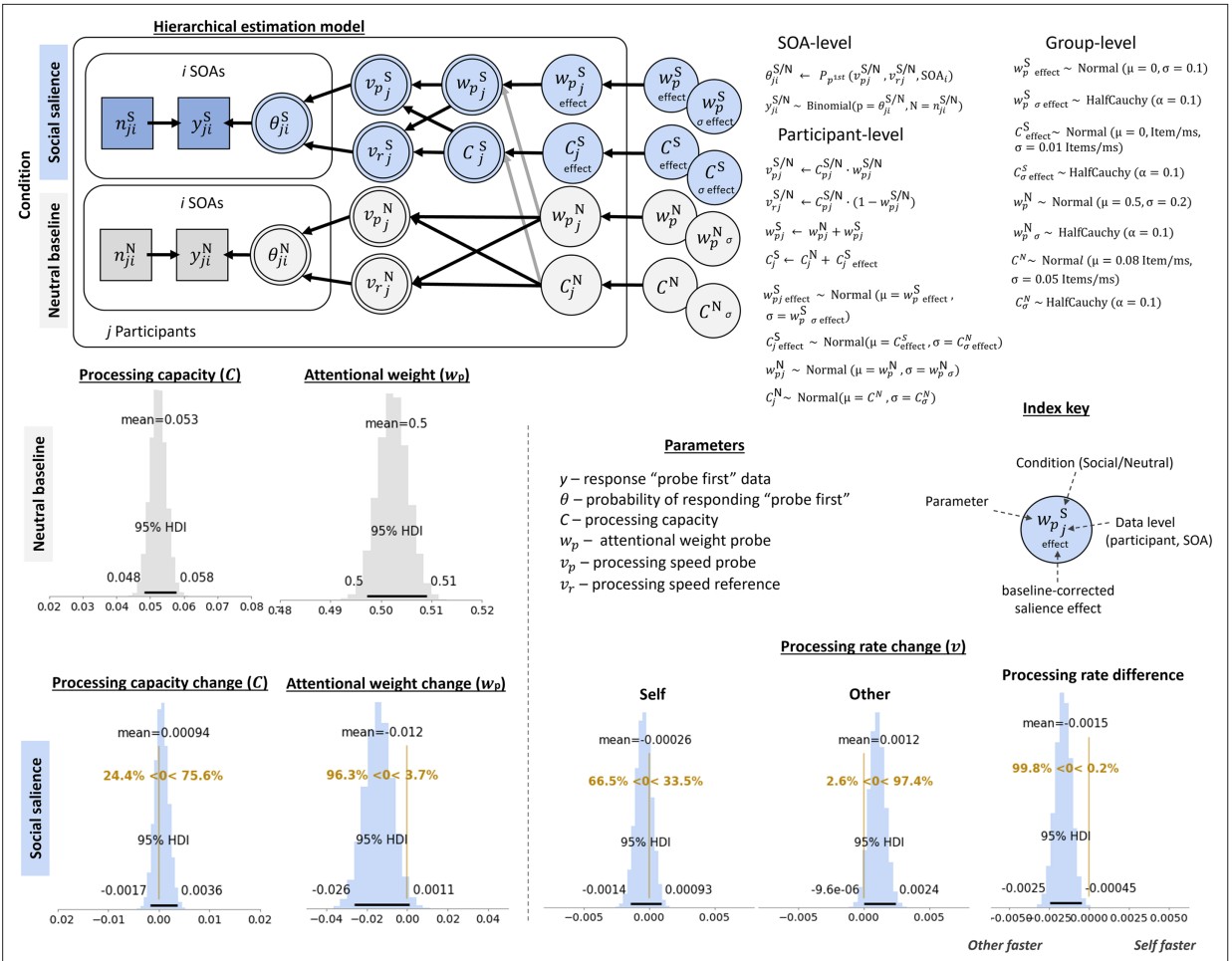

**Figure 4.** Model structure and cross-experimental social salience effects. Hierarchical model structure shows how the parameters of interest ($w_{peffect}$, $C_{effect}$, $\Delta v_p$, and $\Delta v_r$) were estimated from within-participant differences between the neutral baseline (gray) and social salience (blue) condition. The better model is depicted, in which processing capacity was estimated for each condition separately, suggesting that changes in absolute processing rates, rather than relative attentional weights, have been underlying attentional selection effects of social salience. Mathematical formalization of the relation between the model nodes is given on the right. Density plots indicate the highest density estimates for the different processing parameters of interest. Neutral baseline parameters are given in absolute parameter values, with processing capacity shown as items/ms and the relative attentional weight for the probe (a shape that was subsequently associated with the self). Social salience parameters are shown in change scores, relative to baseline, depicting an increase and decrease in processing capacity and attentional weight, respectively. Absolute processing rate changes for the probe (self-associated) and reference (other-associated) shapes, as well as their relative change, are shown on the bottom right.

The online version of this article includes the following figure supplement(s) for figure 4:

**Figure supplement 1.** Individual estimates: Absolute processing rates ($v_p$) for the probe stimulus (self-associated), shown for individual participants for the social baseline (gray; $v_{pBaseSoc}$) and the social salience condition in which the identity had to be reported (dark blue; $v_{pSoc}$).

are corrected in the reported parameter values, which is indicated in the change scores ($w_{peffect}$, $C_{effect}$, $\Delta v_p$, and $\Delta v_r$).

To arbitrate between the mechanisms underlying effects of social salience, we compared two models, applied to data from two experiments ($N=140$): one that estimated a single processing capacity parameter across conditions, and one that estimated processing capacity parameters for every single condition. These define different assumptions we may have about the mechanisms underlying the attentional selection of socially relevant information. If more or fewer processing resources are required to encode socially relevant information into visual short-term memory, we would expect a condition-specific change in processing capacity. This should favor the model that estimates condition-specific processing $C$ s. On the other hand, if the same processing resources are used for encoding

and selecting neutral and socially relevant information, but distributed differently, this should be reflected in a better fit of a model with a single $C$ parameter.

We used the leave-one-out (loo) cross-validation Information Criterion (**Vehtari et al., 2017**), which accounts for model complexity, to compare a model with condition-specific $C$ parameters vs a model with a single, condition-unspecific $C$ parameter. This comparison showed that the best estimated model used condition-specific $C$ parameters ($\Delta loo$ = 14.2; $\Delta se$ = 6.41; $weight_{indiv}$ = 0.86, which can loosely be interpreted as the probability of the model compared to the other). Consequently, social salience introduced not only a change in attentional weights across the perceptual objects, but also a change in processing capacity (see **Figure 1c**). Hence, changes in absolute processing rates, rather than relative attentional weights, are underlying attentional selection effects of social salience. The full model structure of the better model is depicted in **Figure 4**. Note that changes in absolute processing rates, relative to baseline, were estimated for each of the perceptual objects separately: the self-associated shape, defined as the probe ($\Delta v_p$), and the other-associated shape, defined as the reference ($\Delta v_r$).

Inspecting the impact of social association on processing rates, relative to the neutral baseline condition, showed a relative processing advantage for the other-associated stimulus (i.e. $\Delta v_r > \Delta v_p$). That is, surprisingly, the other-associated stimulus was processed 1.2 [$HDI^{95}$: 0 to 2.4] Hz faster after social association, while the self-associated stimulus was processed 0.26 [$HDI^{95}$: –1.4 to 0.93] Hz slower. Relatively, there was an advantage of the other-associated stimulus over the self-associated stimulus of 1.5 [$HDI^{95}$: 0.45 to 2.5] Hz.

Across perceptual objects, there was a slight increase in processing capacity $C$ from 53 Hz to 54 Hz; however, the evidence for a substantial increase was not strong (75.6% of the HDI suggested an increase in processing capacity; **Figure 4**). The mean of the relative attentional weight of the self-associated perceptual object was 0.50 [$HDI^{95}$: 0.50 to 0.51] at baseline, suggesting there were no strong object-specific biases at the group level. After social association, there was a decrease in relative attentional weight attributed to the self-associated stimulus to 0.488 [$HDI^{95}$: 0.474 to 0.498].

This cross-experimental parameter inspection revealed that participants exhibited an attentional selection bias toward socially associated information. Interestingly, enhanced processing speed was observed for other-associated rather than self-associated information, a pattern that diverged from our prediction.

## Does self-relatedness bias attentional selection automatically or does it require explicit social decoding? (Experiment 1)

Changing the decisional dimension, either requiring explicit decoding of the social associations, or merely the perceptual features, allowed testing whether social relevance biases attentional selection automatically at perceptual feature representations or whether it requires explicit decoding of the social identity. The differentiation between these mechanisms holds the inherent assumption that automatic alterations in the perceptual feature representation are faster, while active decoding of the social identity requires more processing time. Indeed, an exploratory analysis on reaction time data showed that participants responded significantly quicker in the perceptual decision dimension (RT$\mu$: 763.91 [$CI^{95}$: 729 to 799]ms) than in the social decision dimension (RT$\mu$: 897.33 [$CI^{95}$: 850 to 944]ms; $BF_{10} = 1.064^{1011}$). Note that the order in which the perceptual decision dimension and social decision dimensions were tested was counterbalanced across participants, ruling out the possibility that the reaction time difference was merely a training effect. Instead, this may suggest that additional processing stages were involved in the condition that required active decoding of the social identity association.

To assess whether, in line with the cross-experimental analysis, processing capacity changes contributed to explaining the attentional selection of socially relevant information, we conducted model comparisons on all conditions included in this experiment: neutral baseline, social salience with the perceptual decision, social salience with the social decision. In Experiment 1 (*N*=69), the condition-specific individual $C$ model (**Figure 5**) was favored ($\Delta loo$ = 7.65; $\Delta se$ = 6.07; $weight_{indiv}$ = 0.71). This, as above, suggested that social salience introduced changes in relative attentional weights and processing capacity, hence, in the absolute processing rates.

Relative to the neutral baseline condition, self-associated stimuli showed increased absolute processing rates compared to other-associated stimuli (i.e. $\Delta v_p - \Delta v_r$). However, this was only the case

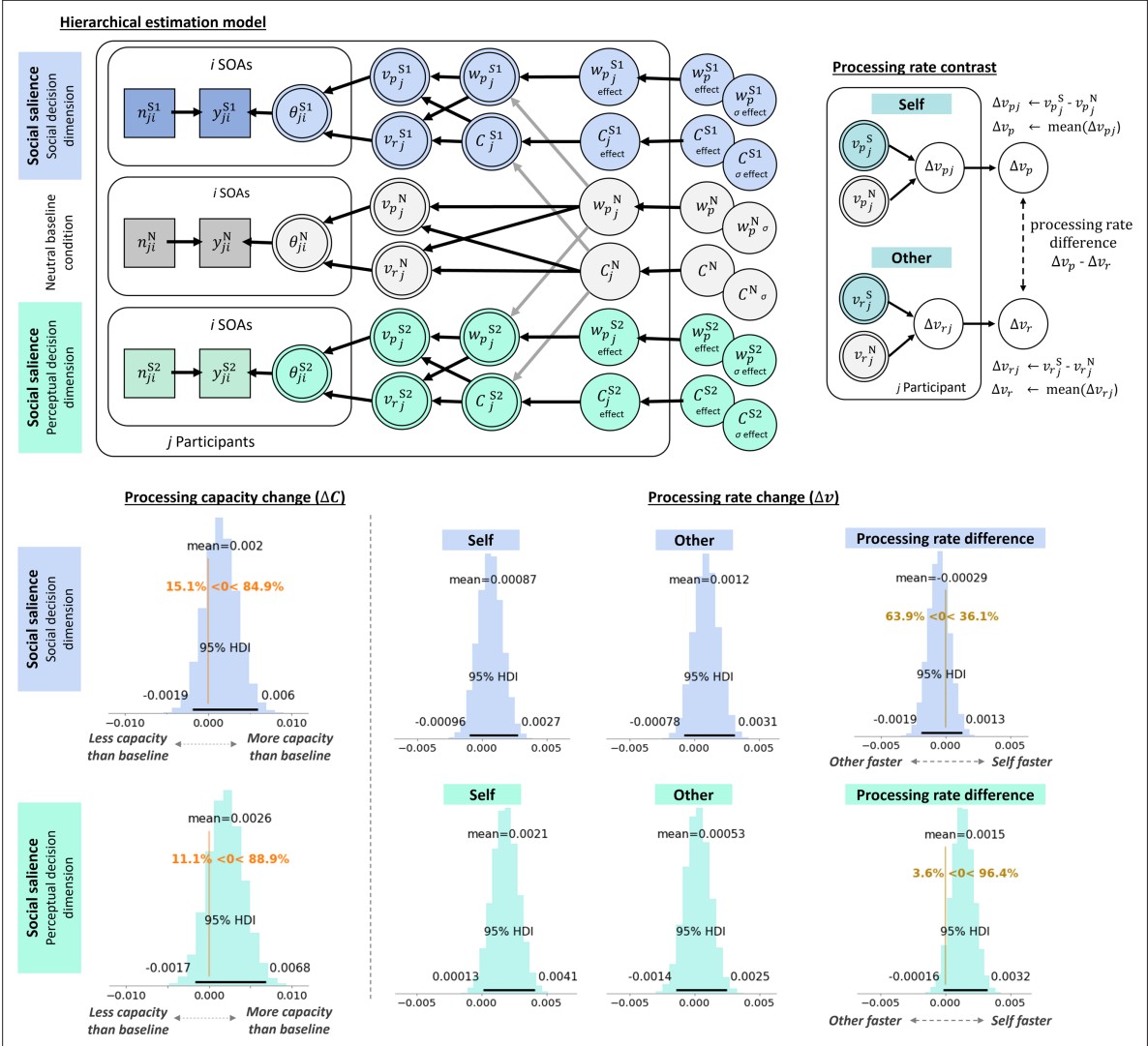

**Figure 5.** Social salience effects with different decision dimensions. Hierarchical model structure shows how social salience effects were estimated between social salience conditions (blue: social decision dimension; turquoise: perceptual decision dimension) and the neutral baseline condition (gray; see *Figure 4* caption for details and formalizations) in Experiment 1. The right plot shows how relative processing rates were calculated, at the individual participant level, from social salience-induced processing rate changes for the self-associated and other-associated shapes. Density plots indicate the group-level highest density intervals for the processing capacity and absolute processing rate estimates, given in items/ms. Additionally, the 95% HDIs are presented alongside the group means. The relative change in processing rates ($\Delta v_p - \Delta v_r$) can be interpreted directly as the processing rate advantage of the self-associated over the other-associated stimuli. Raw response data and parameter estimates for individual participants are provided in *Figure 5—figure supplement 1 and 2*, respectively.

The online version of this article includes the following figure supplement(s) for figure 5:

**Figure supplement 1.** Individual psychometric functions: Individual participant response data indicating the proportion with which participants responded that the probe flickered first as a function of stimulus onset (flicker) asynchrony.

**Figure supplement 2.** Individual estimates: absolute processing rates ($v_p$) for the probe stimulus (self-associated), shown for individual participants in Experiment 1.

when participants were asked to report which perceptual object flickered first, that is in the perceptual decision dimension (*Figure 5*). Here, the self-associated stimulus was processed 1.5 [$HDI^{95}$: –0.16 to 3.2] Hz faster than the other-associated stimulus. This was driven by an overall increase in processing capacity by 2.6 [$HDI^{95}$: –1.7 to 6.8] Hz, specifically for the self-associated stimulus (2.1 [$HDI^{95}$: 0.13 to

4.1] Hz). The processing rate for the other-associated stimulus did not show a consistent increase (0.53 [$HDI^{95}$: –1.4 to 2.5] Hz).

When participants had to report whose shape flickered first based on the social association, there was no advantage of the self-associated stimulus (0.87 [$HDI^{95}$: –0.96 to 2.7] Hz; *Figure 5*). Instead, the Bayesian parameter estimates provided more evidence in favor of the other-associated stimulus (1.2 [$HDI^{95}$: –0.78 to 3.1] Hz) being processed faster (0.29 [$HDI^{95}$: –1.9 to 1.3] Hz), in line with the cross-experimental analysis (63.9% of the HDI suggested an advantage of the other-associated stimulus).

Results from experiment 2 demonstrated a faster, more automatic attentional selection for self-associated information when the decision did not require explicit social decoding. When the social identity had to be judged, processing speed for self-associated information decreased. Contrary to the hypothesis that social decoding is necessary for self-prioritization to emerge, these findings suggest that attentional selection can operate automatically to prioritize self-associated information.

## Does self-association affect attentional selection in a similar way to perceptual salience, and how do social and perceptual salience interact with each other? (Experiment 2)

To investigate the similarities and interactions between social and perceptual salience in attentional selection, we asked participants ($N$=71) to complete the TOJ task with socially salient (shape) and perceptually salient (local color) stimuli separately, as well as together. As such, there were four conditions of interest: mere social salience (social decision dimension), mere perceptual salience, social + perceptual salience (self perceptually salient), social + perceptual salience (other perceptually salient). Similar to the previous experiment, participants completed a baseline task to reduce the effects of any participant-specific pre-existing biases.

Note that the definition of the probe differed between the mere perceptual and mere social salience conditions and required the neutral baseline condition to be coded accordingly. This did not affect the stimulus presentation in any way, but merely the interpretation of the 'probe': in the social salience condition and the respective neutral baseline condition, the probe was always a specific shape. In the perceptual salience condition and the respective neutral baseline condition, the probe was randomly defined as one of the two shapes on each trial. This is because, in the perceptual salience condition, the local color that feature induced salience had a random chance of affecting each shape. As such, the probe in the baseline was always defined in such a way that it provided a direct comparison for the respective salience condition. To quantitatively assess the interaction of perceptual and social salience, the effect of perceptual salience was measured with the same defined probe as the perceptual salience condition; however, after shapes had been associated with social identities. This allowed us to draw direct comparisons between the effects of perceptual salience on self- versus other-associated perceptual objects.

In contrast to the previous experiment, the single-$C$ model was favored in this experiment ($\Delta loo$ = 33.0; $\Delta se$ = 9.96; $weight_{single}$ = 0.84). Notably, while this affects the interpretation of underlying mechanisms of social and perceptual salience manipulations together, we report changes in processing rates for consistency and comparability. As absolute processing rates indicate the product of processing capacity and relative attentional weight, a non-meaningful change in processing capacity does not alter the interpretation of processing rates. Hence, changes in relative processing rates (self/other, salient/non-salient) can be interpreted in the same way between experiments.

Across conditions, there was a slight decrease in processing capacity: mere social salience: –1.3 [$HDI^{95}$: –5.3 to 0.28] Hz; mere perceptual salience: –0.92 [$HDI^{95}$: –3.9 to 2.1] Hz; social + perceptual salience (self): –1.9 [$HDI^{95}$: –5.5 to 1.8] Hz; social + perceptual salience (other): –1.1 [$HDI^{95}$: –5.6 to 3.5] Hz.

As indicated in the cross-experimental analysis, the processing rates for other-associated stimuli showed a relative processing advantage compared to self-associated stimuli (i.e. $\Delta v_r > \Delta v_p > \Delta v_p$; –1.6 [$HDI^{95}$: –3 to –0.26] Hz), relative to the neutral baseline condition (*Figure 6*, blue). Here, the processing rate of the self-associated stimulus decreased by 1.3 [$HDI^{95}$: –5.3 to 0.28] Hz, while the processing rate of the other-associated stimulus stayed approximately the same, with a 0.17 [$HDI^{95}$: –1.6 to 1.9] Hz increase.

Assessing the effects of mere perceptual salience on processing suggested an increase in the salient stimulus, relative to the non-salient stimulus (i.e. $\Delta v_p > \Delta v_r$; 6 [$HDI^{95}$: 4.6 to 7.3] Hz; *Figure 6*,

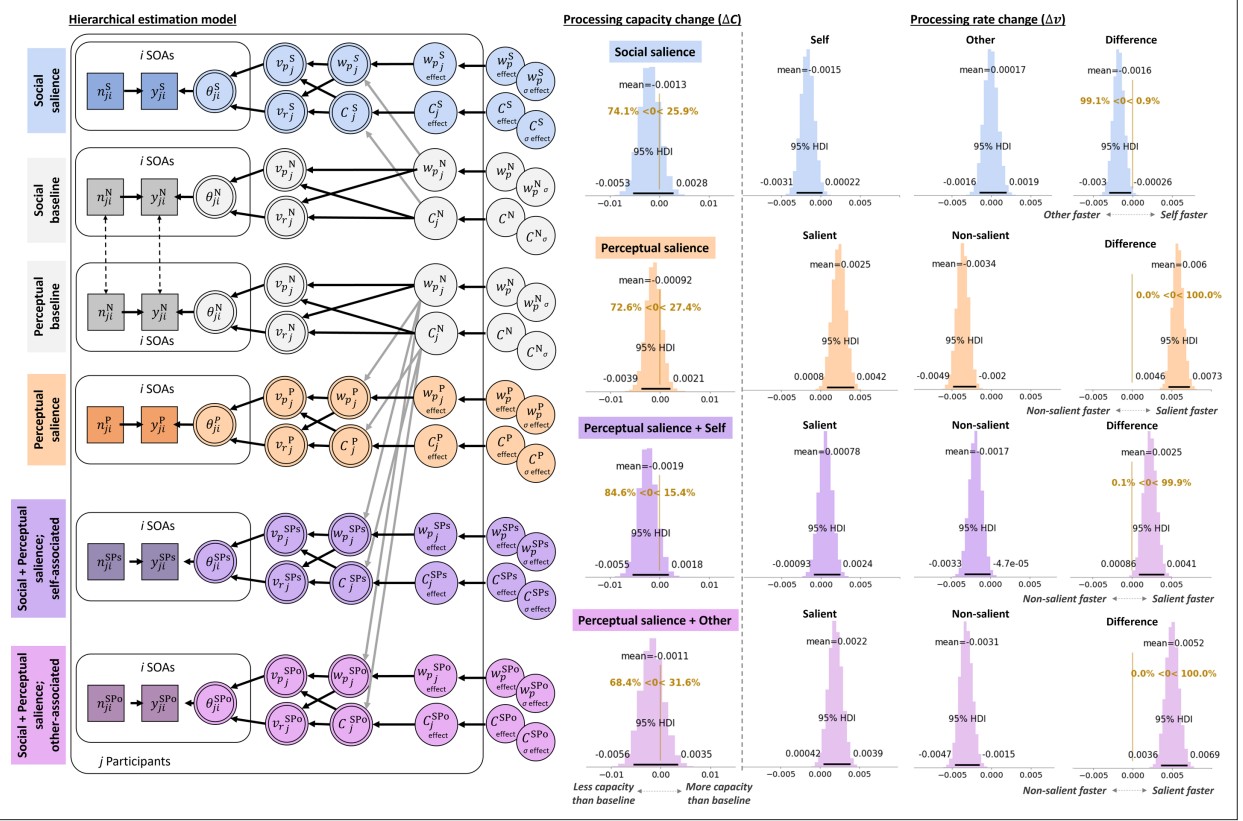

**Figure 6.** Social and perceptual salience effects. Hierarchical model structure shows how social salience only (blue) and perceptual salience only (orange) effects were estimated relative to their respective neutral baseline conditions (gray; see **Figure 4** caption for formalizations and main text for details) in Experiment 2. To assess processing parameters for the interaction of social and perceptual salience (purple, pink), we calculated change scores, indicative of perceptual salience effects, relative to the perceptual neutral baseline. This can be interpreted as the effect of perceptual salience that is present when shapes are associated with the own or another social identity. Density plots indicate the group-level highest density intervals for the processing capacity and processing speed estimates, given in items/ms. Additionally, the 95% HDIs are presented alongside the group means. The relative speed change ($\Delta v_p - \Delta v_r$) can be interpreted directly as the processing rate advantage of the self-associated over the other-associated stimuli (social salience only), or the perceptually salient over the perceptually non-salient stimuli (perceptual salience only, social + perceptual salience). Raw response data and parameter estimates for individual participants are provided in **Figure 5—figure supplements 1 and 2**, respectively.

The online version of this article includes the following figure supplement(s) for figure 6:

**Figure supplement 1.** Individual psychometric functions: Individual participant response data indicating the proportion with which participants responded that the probe flickered first as a function of stimulus onset (flicker) asynchrony.

**Figure supplement 2.** Individual estimates: Absolute processing rates ($v_p$) for the probe stimulus (perceptually salient), shown for individual participants for the baseline (gray; $v_{pBase}$), for the mere perceptual salience condition (orange; $v_{pPerc}$), for the perceptual salience condition in which the probe was self-associated (purple; $v_{pPerSelf}$), and the perceptual salience condition in which the probe was other-associated (pink; $v_{pPerOther}$).

orange). This was driven both by an increase in processing rates of the salient (2.5 [$HDI^{95}$: 0.8 to 4.2] Hz) and a decrease in processing rates of the non-salient (–3.4 [$HDI^{95}$: –4.9 to –2] Hz) stimuli.

Taken together, as also confirmed in the cross-experimental analysis, attentional selection favored the other-related information when social identity had to be judged. In contrast, perceptual salience, as predicted, led to increased processing speed for the more salient stimulus.

Presenting stimuli that were both self-associated and perceptually salient reduced the relative processing rate advantage of perceptual salience (**Figure 6**, purple). That is, while there was still a processing rate increase of perceptually salient, self-associated stimuli (2.5 [$HDI^{95}$: 0.86 to 4.1] Hz), it was much smaller compared to perceptually salient, non-associated stimuli (6 [$HDI^{95}$: 4.6 to 7.3] Hz) or other-associated stimuli (5.2 [$HDI^{95}$: 3.6 to 6.9] Hz; **Figure 6**, pink). This was specifically driven by a smaller processing rate increase towards the salient, self-associated stimulus (0.78 [$HDI^{95}$: –0.93 to 2.4]), but also less suppression of the non-salient stimulus, suggesting that self-relatedness and perceptual salience interacted somewhere along the processing hierarchy.

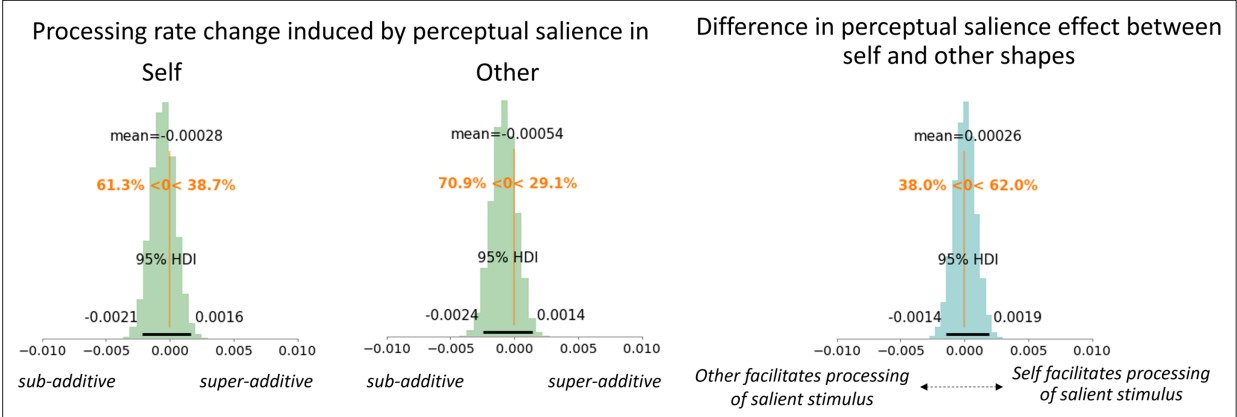

**Figure 7.** Interaction of social and perceptual salience. Interaction effects of social and perceptual salience on processing rates. Probability density plots of processing rate change parameters when the stimulus was perceptually salient and self-associated (left panel) or other-associated (middle panel). The right panel shows the difference in perceptual-salience induced processing benefit between the self- and other-associated stimuli.

The online version of this article includes the following figure supplement(s) for figure 7:

**Figure supplement 1.** Formalization of interaction: formalization of interaction effect assessment, using attentional weights.

To elucidate more directly whether perceptual and social salience are processed independently or interact, we assessed whether the effects of social and perceptual salience were additive, or whether there were deviations in processing rates when social and perceptual salience were present simultaneously. To that end, we added the processing rate changes resulting from perceptual salience alone and social associations alone and subtracted them from the processing rate changes in the condition in which perceptual salience and social associations were presented together (see *Figure 7—figure supplement 1*). In the salient stimulus, there were no systematic deviations between perceptual salience alone and perceptual salience when it was presented on a self-related stimulus (–0.28 [$HDI^{95}$: –2.1 to 1.6] Hz; see *Figure 7*) or an other-related stimulus (–0.54 [$HDI^{95}$: –2.4 to 1.4] Hz).

Next, we quantified the differences in perceptual salience effects between self-associated and other-associated stimuli, taking the individual social and perceptual only effects into account. Parameter inspection of the processing rates suggested that stimuli were not systematically more or less attended when perceptually salient stimuli were self-related compared to other-related (0.26 [$HDI^{95}$: –1.4 to 1.9] Hz). This indicates an additive effect, which would be expected if social and perceptual salience were processed independently.

## Can self-relatedness effects in attentional selection explain self-prioritization in perceptual matching?

To ascertain that participants learned the correct association and exhibited a typical SPE, they completed a common shape-label matching task (*Sui et al., 2012*). Across both experiments, strong SPEs were present, indicated by an enhanced accuracy towards match-trials of self-associated, compared to other-associated, stimuli (see *Figure 8*; Experiment 1: $\delta$ = –1.064 [$CI^{95}$: –1.38 to –0.75], $BF_{10}$ = 3.23$^{109}$; Experiment 2: $\delta$ = –0.982 [$CI^{95}$: –1.20 to –0.77], $BF_{10}$ = 4.47$^{1017}$).

Interestingly, while self-associated stimuli were prioritized in shape-label matching, they showed reduced processing rates in attentional selection (as outlined above), at least when the decisional dimension was in the social domain. To assess the degree to which social biases in early attentional selection contribute to the expression of SPEs in perceptual matching, we correlated the differences in processing rate differences between the self and other associated stimulus, that is the self-induced change in processing rates, with the SPE magnitude. Note that, as SPEs measured in the matching paradigm are likely the cumulative result of self-biases at different levels of processing, we would not expect predictive effects to be particularly strong.

In Experiment 1, the individual SPE benefits in the shape-label matching task correlated with the socially-induced processing rate changes in the social decision dimension ($r$=0.354 [$CI^{95}$: 0.11 to 0.54]; $BF_{10}$ = 8.23; see *Figure 9*), but not in the perceptual decision dimension ($r$=0.069 [$CI^{95}$: –0.18 to 0.31]; $BF_{10}$ = 0.181). In Experiment 2, the individual SPE benefits in the shape-label matching task negatively

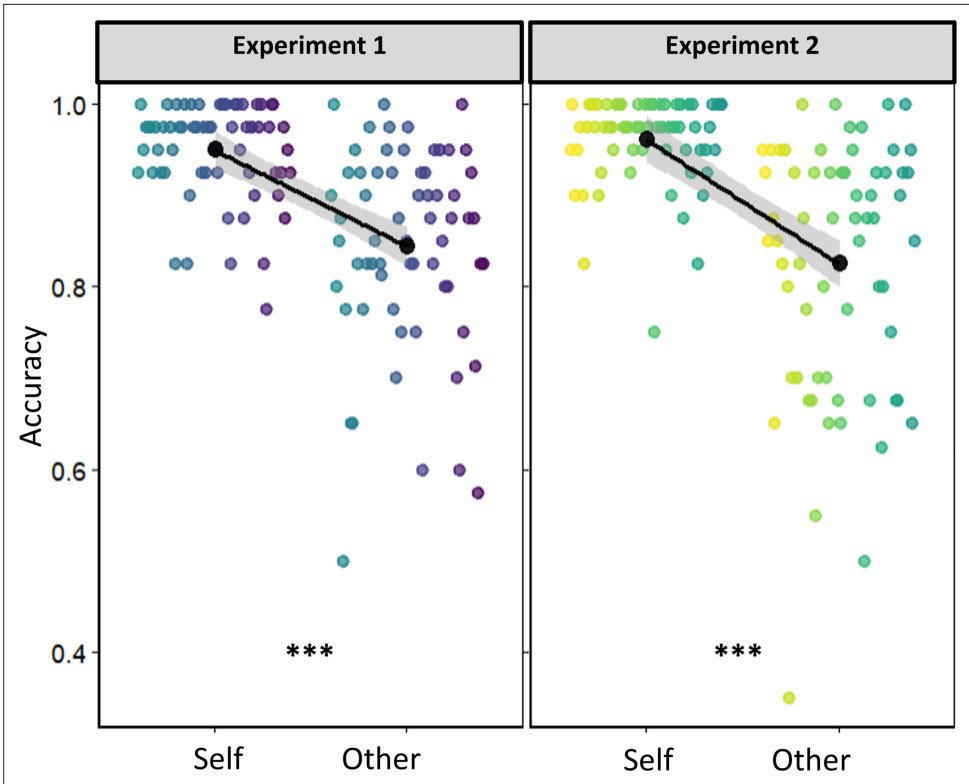

**Figure 8.** Self-prioritization in matching. Matching task results showing robust self-prioritization effects in both experiments, indicated by the enhanced accuracy towards self-associated information compared to other-associated information on match-trials. Colored points indicate individual participants. Black superimposed points and error bands indicate group means and $CI^{95}$.

correlated with the socially induced processing rate changes in the perceptual decision dimension ($r=-0.328$ [$CI^{95}$: –0.52 to –0.08]; $BF_{10}$ = 4.53; see **Figure 9**), but not the processing rate changes in the social decision dimension ($r=-0.138$ [$CI^{95}$: –0.37 to 0.11]; $BF_{10}$ = 0.278).

Pooling data across experiments indicated no correlations between the SPE benefits and processing rate changes obtained either through perceptual or social decision dimensions (perceptual: $r = -0.087$ [$CI^{95}$: –0.25 to 0.09]; $BF_{10}$ = 0.177; social: $r = -0.029$ [$CI^{95}$: –0.20 to 0.15]; $BF_{10}$ = 0.117). However, it showed that the processing rate changes in the social decision dimension were negatively correlated with the processing rate changes in the perceptual decision dimension (processing rates  $r = -0.243$ [$CI^{95}$: –0.39 to –0.08]), $BF_{10}$ = 6.58; see **Figure 9**, presented with individual experiment data super-imposed, which was driven by a differential allocation of relative attentional weights $w$ ($r=-0.268$ [$CI^{95}$: –0.41 to –0.11]; $BF_{10}$ = 16.93). In other words, individuals who showed a faster selection of (socially or perceptually) salient information at the perceptual shape feature level showed the least processing facilitation towards self-associated shapes when reporting the social association. At the same time, those who showed a faster selection of perceptually non-salient or other-associated information when reporting the perceptual feature showed the strongest attentional facilitation for self-associated shapes when reporting the social association. This suggests a trade-off in attentional selection between the two decisional dimensions, leading to opposing effects.

Lastly, we explored the contribution of perceptual and social salience to attentional selection of information that is both perceptually and socially salient in Experiment 2 via the correlation of shifts in relative attentional weights. This indicated that perceptual salience effects in other-associated stimuli were more strongly correlated with the mere perceptual-salience effects ($r=0.532$ [$CI^{95}$: 0.33 to 0.67]; $BF_{10}$ = 10317.1) rather than the mere social-salience effects ($r=-0.285$ [$CI^{95}$: –0.48 to –0.05]; $BF_{10}$ = 2.56). In contrast to that, perceptual salience effects in self-associated stimuli were more strongly correlated with mere social salience effects ($r=0.429$ [$CI^{95}$: 0.21 to 0.59]; $BF_{10}$ = 137.87)

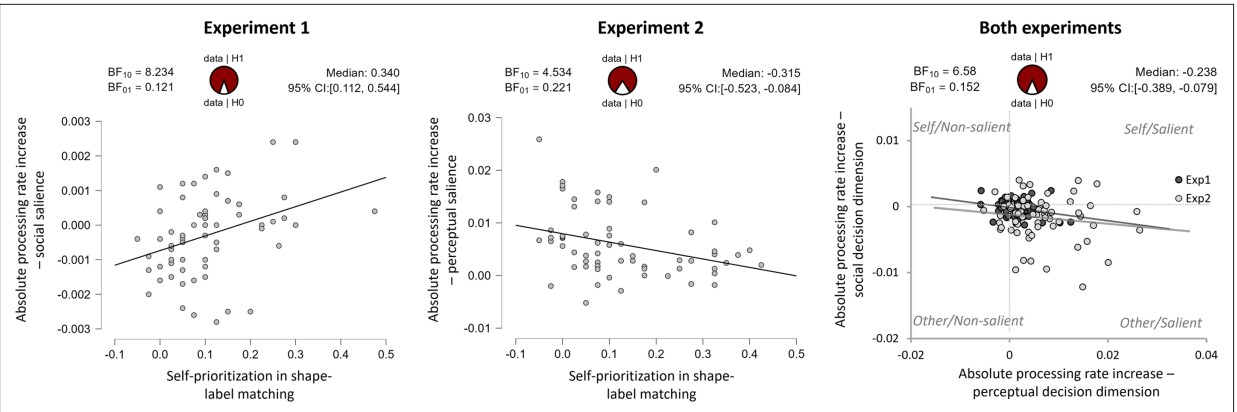

**Figure 9.** Individual differences reveal how task parameters are related. Scatter plots for Experiments 1 and 2 showing individual self-prioritization effect benefits in the shape-label matching task predicting socially induced changes in attentional processing rates for the self-associated/salient (positive) or other-associated/non-salient (negative) stimuli. Solid lines indicate linear best fit. The right scatter plot shows the changes in individual, absolute processing rates in response to social versus perceptual decision judgements, for both experiments. Experiments and respective best linear prediction lines are color-coded in darker (Experiment 1) and lighter (Experiment 2) gray, to allow distinguishing between conditions that used social salience (dark gray) or perceptual salience (light gray) with the perceptual decision dimension. Bayes factors assessing the probability of a linear correlation and posterior estimation info is provided above each plot.

than mere perceptual salience effects ($r$=0.31 [$CI^{95}$: 0.08 to 0.50]; $BF_{10}$ = 4.45). Bayesian regression model selection supported this: While a model that included perceptual and social salience effects together explained the perceptual salience effects in self- and other-associated stimuli best (Other: $P(M|Data)$ = 0.504; $BF_M$ = 4.07; $R^2$ = 0.328; Self: $P(M|Data)$ = 0.682; $BF_M$ = 8.60; $R^2$ = 0.324), inspection of the posteriors indicated that, for perceptually salient, other-associated shapes, mere perceptual salience effects were the best predictor ($BF_{inclusion}$=4638.74; vs social: $BF_{inclusion}$ = 2.83; *Table 1*). For self-associated shapes, on the other hand, mere social salience effects were the best predictor ($BF_{inclusion}$=2458.52) while a susceptibility to mere perceptual salience also contributed ($BF_{inclusion}$=153.25; *Table 2*). In other words, when information was not self-related, attentional selection was mostly driven by perceptual salience. In contrast, when information was related to the self, attentional selection was driven more strongly by social than perceptual salience.

## Discussion

The present study investigated the mechanisms by which social salience biases attentional selection. Using a theory-informed Bayesian modeling framework of visual attention (TVA), we established evidence for effects of social and perceptual salience and assessed their interaction. In conjunction with the TOJ paradigm, TVA offers a clearly formalized, systematic framework for understanding attentional selection, allowing researchers to probe mechanisms via which attention facilitates encoding of sensory information into visual short-term memory. Its reliability and theoretical importance in describing attention and attentional selection of perceptually salient information have been demonstrated in previous studies (*Krüger et al., 2016*; *Krüger et al., 2017*). The present study

**Table 1.** Posterior coefficient summaries for perceptual salience and other-association.

| Coefficient | P(incl) | P(incl\|Data) | $BF_{inclusion}$ | Mean | SD | $CI^{95}$ Lower | $CI^{95}$ Upper |
|---|---|---|---|---|---|---|---|
| Intercept | 1.000 | 1.000 | 1.0 | 0.052 | 0.011 | 0.031 | 0.073 |
| $\Delta w_p^{Per}$ | 0.556 | 1.000 | 4638.74 | 0.833 | 0.200 | 0.434 | 1.232 |
| $\Delta w_p^{Soc}$ | 0.556 | 0.779 | 2.83 | −1.191 | 0.624 | −2.435 | 0.053 |
| $\Delta w_p^{Per*} \Delta w_p^{Soc}$ | 0.333 | 0.414 | 1.42 | 6.081 | 6.667 | −7.216 | 19.378 |

**Table 2.** Posterior coefficient summaries for perceptual salience and self-association.

| Coefficient | P(incl) | P(incl|Data) | $BF_{inclusion}$ | Mean | SD | $CI^{95}$ Lower | $CI^{95}$ Upper |
|---|---|---|---|---|---|---|---|
| Intercept | 1.000 | 1.000 | 1.0 | 0.021 | 0.006 | 0.008 | 0.034 |
| $\Delta w_p^{Per}$ | 0.556 | 0.995 | 153.25 | 0.405 | 0.119 | 0.167 | 0.643 |
| $\Delta w_p^{Soc}$ | 0.556 | 1.000 | 2458.52 | 0.630 | 0.372 | –0.111 | 1.372 |
| $\Delta w_p^{Per} \star$ $\Delta w_p^{Soc}$ | 0.333 | 0.573 | 2.68 | 4.323 | 3.976 | –3.606 | 12.252 |

demonstrates the applicability of TVA in examining attentional selection of socially salient information, and its interaction with perceptual salience.

## Social salience effects in attentional selection – processing levels (Experiment 1)

For the first time, we established that mere social relevance influences how attentional resources are allocated across the visual field: self-reference leads to changes in processing capacity and the allocation of resources across the visual field. That is, when sensory information becomes associated with social identities, its social connotation affects early attentional selection. Interestingly, the active decoding of social information was not necessary for this effect to take place. When participants had to decide which of two shapes flickered first, processing rates for self-associated shapes increased, relative to other-associated shapes, as a result of social association (i.e. relative to baseline). The conscious decoding of social associations was unnecessary for this effect to emerge, providing evidence against the claim that self-relatedness strictly has to be a conscious, goal-related feature in order to induce self-prioritization (e.g. *Golubickis and Macrae, 2023*; *Woźniak and Knoblich, 2022*). Instead, it supports further evidence showing that self-prioritization may emerge from the intrinsic nature of self-processing (*Lee et al., 2021*; *Zhang et al., 2023*) and unfolds across different stages of the processing hierarchy (*Desebrock and Spence, 2021*; *Reuther and Chakravarthi, 2017*; *Scheller and Sui, 2022b*), with early processing stages being affected in an almost automatic fashion (*Alexopoulos et al., 2012*; *Geng and Xu, 2011*; *Humphreys and Sui, 2015*; *Sui et al., 2014*; *Yin et al., 2019*). It further supports the framework of the Self-Attention Network (SAN; *Humphreys and Sui, 2016*), which outlines the crucial role of attention in the behavioral facilitation of self-related information. This framework suggests that early self-prioritization arises in an automatic fashion while subsequent, active suppression is required when non-self-related information becomes goal-relevant.

Further evidence for the multi-stage nature of self-prioritization in information processing is given by the fact that SPEs, measured via the shape-label matching tasks (*Sui et al., 2012*), which involves several higher-level processing stages, only correlated with the processing rate changes at higher processing levels in the current TVA-TOJ task. In other words, the SPEs measured via shape-label matching are more similar to the individuals' attentional selection effects at higher-level, social decoding stages. Note, however, that self-relatedness biases at later processing levels do not rule out its automatic capture of attention at earlier levels of perceptual processing.

Previous studies that established that self-ownership can favor attentional selection (*Constable et al., 2019*; *Jublie and Kumar, 2021*; *Truong et al., 2017*) suggested that a social decision dimension was necessary (*Constable et al., 2019*). The present investigation, therefore, contrasted perceptual and social salience across corresponding decisional dimensions: shape-specific (perceptual) versus identity-specific (social) decisions. Interestingly, we found that the decisional dimension had differential effects on social salience effects. Automatic biases that were favoring the self-associated stimulus were only present at the perceptual decision level, but not at the social decision level. When active decoding of social information was required, the same participant group that showed self-prioritization in early attentional selection instead showed no bias for the self, but a bias towards other-associated shapes. However, the relative slowing in processing rates for the self-associated stimuli was individual-specific. That is, those with the strongest self-relatedness benefit in shape-label matching showed the smallest decrease/the largest increase in processing rates for the self. This

suggests that, rather than a general self-bias in the population, the benefit depended on the strength of self-representation. In contrast to previous studies, we controlled for any individual-specific, pre-existing biases, such as shape-preference biases, by including a baseline task prior to social association induction. This control allowed us to ensure that the attentional biases we measured were the direct result of perceptual and social salience.

While we observed automatic attentional allocation effects towards the self-associated stimulus, the explicit decoding of the associated social identity led to a relative slowing of processing rates for the self-associated stimulus. This is opposite to what we expected and what was reported in previous studies (e.g. *Constable et al., 2019*; *Jublie and Kumar, 2021*), begging the question as to why this pattern emerged in the present task. One possibility is the type of social association that was being used: mere social association with arbitrary objects versus ownership (*Constable et al., 2019*) or bodily self-representation (faces; *Jublie and Kumar, 2021*). Another alternative may be that the present study used a different type of event, for which the temporal order had to be established: Instead of assessing the order of stimulus onset, we asked participants to determine the order in which two stimuli flickered. We chose this particular event as TVA posits that attentional selection requires an initial wave of unselective attentional capacity buildup across the visual field, before attentional weights are selectively distributed across the processing channels to alter the rate at which information is subsequently processed (*Bundesen et al., 2005*). Even bottom-up driven perceptual salience effects require an initial 150–200 ms to build up sufficient attentional processing capacity to show the typical, beneficial perceptual salience effects (*Krüger and Scharlau, 2021a*). A previous series of experiments on perceptual salience effects using TVA-TOJ showed that other events such as stimulus onset and stimulus offset failed to elicit expected benefit of pure bottom-up perceptual salience manipulations (orientation, color), likely because they induced more permanent changes in the salience of the display (*Krüger et al., 2016*). Instead, a brief stimulus flicker allowed to probe the mechanisms of perceptual salience. Together with our findings of the differential effects of decision dimension in Experiment 1, this suggests that previous reports of prior entry (stimulus onset, social decision dimension in *Constable et al., 2019*; *Truong et al., 2017*) likely resulted from higher-level processing stages. Hence, the present study provides the first account of subconscious, automatic effects of mere self-relatedness on early attentional selection during perception. The factors that lead to changes in directionality of these effects remain to be explored in future studies.

## Comparing social and perceptual salience (Experiment 2)

By combining social associations with perceptual salience, the present study allowed establishing, for the first time, similarities, differences, and interactions between social and perceptual salience effects. Experiment 2 revealed that changing local color features of the targets, relative to the background objects, produced the expected perceptual-salience effects, which have been reported in previous studies using TVA-TOJ (*Krüger et al., 2016*; *Krüger et al., 2017*). That is, local color alterations produced a shift in attentional weights towards the more salient stimulus. The increase in attentional weight in the present study $w_{peffect}$ = 0.059 [0.037 – 0.081] was similar, albeit slightly larger, in magnitude to the one reported by *Krüger et al., 2016*; $w_{peffect}$ = 0.043. Inducing social salience, on the other hand, led to decreases in processing rates for the self-associated (presumably salient) stimulus, at least when active decoding of the social identities was required – an unpredicted effect possibly reflecting later, compensatory mechanisms overriding automatic perceptual benefits obtained by self-relatedness (as outlined above).

We investigated whether the different decisional dimensions used for social salience versus perceptual salience contributed to any effect differences, by including two conditions assessing the effects of perceptual salience on specifically associated shapes (self, other) under the same decisional dimension. Here, we observed that perceptual salience effects were diminished by social associations, with a stronger effect reduction in the self-associated shape. Notably, the reduction in perceptual salience benefit corresponded to the decrease in processing rates for the self-associated and other-associated shapes, suggesting additive rather than interactive effects of perceptual and social salience. This additive pattern may reflect that perceptual and social salience were induced in different features (color and shape, respectively) and may therefore have led to a trade-off in attentional resource allocation. Indeed, even perceptual salience effects across two different feature dimensions, luminance and orientation contrasts, have previously been reported to be additive (*Krüger et al., 2017*), reflected in

the TVA parameter Kappa (*Nordfang et al., 2013*). The processing speeds at which this information races can be influenced both by social and perceptual salience. While not completely ruling out that perceptual and social salience interact at any point in the processing hierarchy, our findings indicate that, when they are induced in different stimulus features, their salience effects on attentional selection are independent and additive. Using TVA's report categories and the modulatory Beta parameter, future work could investigate the interactions between perceptual and social categorizations.

Interestingly, while the average effects of social and perceptual salience were mostly additive, their respective contributions differed: prediction model comparisons revealed that when the self-related stimulus was also perceptually salient, the processing benefit magnitude was mostly predicted by the social salience of the stimulus, and to a lesser, even if substantial, extent by perceptual salience. On the contrary, for other-related stimuli, the processing benefit magnitude was determined primarily by its perceptual salience alone. This suggests that the trade-off between perceptual and social salience in attentional selection differs between self- and other-associated stimuli: Self-relatedness has the power to partially overwrite the effects of bottom-up perceptual salience on attentional selection, while other-association does not.

## Conclusions

Overall, a consistent finding that emerges across the literature is that prioritization of social information requires the deployment of attention (*Alexopoulos et al., 2012*; *Humphreys and Sui, 2016*; *Sui and Rotshtein, 2019*; *Wade and Vickery, 2018*). However, the mechanisms by which attention shapes different stages of information processing to facilitate self-related processing are not fully understood. Here, we show that mere self-association with arbitrary shapes can alter attentional selection at early, perceptual processing stages, leading to increased processing rates for self-associated stimuli. Crucially, this self-prioritization occurs in an automatic fashion that does not require the active decoding of the associated social identities. Secondly, we show that varying the decisional dimension alters the degree of processing benefit obtained from social relevance, highlighting the multi-stage nature of self-relatedness effects in information processing. Thirdly, our results suggest that social and perceptual salience do not interact during information processing but are likely processed in parallel: while perceptual salience showed robust increases in processing rates, this effect reduced in socially associated stimuli, consistent with the processing rate reductions associated with self and others. Lastly, exploratory intra-individual correlations showed that the relative contribution of social and perceptual salience to attentional selection depended on the social association: self-relatedness was considered more strongly than perceptual salience, while perceptual salience was considered more strongly than other-relatedness. This investigation provides the first evidence outlining how self-relatedness leads to automatic, perceptual benefits through early attentional selection, and how social and perceptual salience shape attentional deployment. These findings shed light on core mechanisms underlying the pervasive SPEs that fundamentally shape human information processing hierarchy from early perceptual encoding to higher-order cognition and conscious awareness.

## Methods

### Participants

We recruited 140 adult participants across the two experiments (103 women; age M ± SD = 22.7±4.2 years). Participants were research study volunteers that reported to have normal or corrected-to-normal vision and no developmental or cognitive impairments.

### Sample size justification

The main goal of the present study was to quantify effects of social salience and compare them to perceptual salience effects in a model-based framework. To that end, we conducted a power analysis for detecting the main effects of interest, that is, a change in attentional weights in conditions where targets were socially or perceptually salient. The TOJ-TVA model that was used in the analysis enables a detailed Bayesian power analysis. Note that in the context of Bayesian parameter estimation, there is no concept of 'power' in the classic sense as there is no strict decision threshold (e.g. a p-value) but rather a gradual quantification of the credibility of parameter values. Nevertheless, in order to maximize the chance of obtaining informative results with a high degree of certainty, we employ a

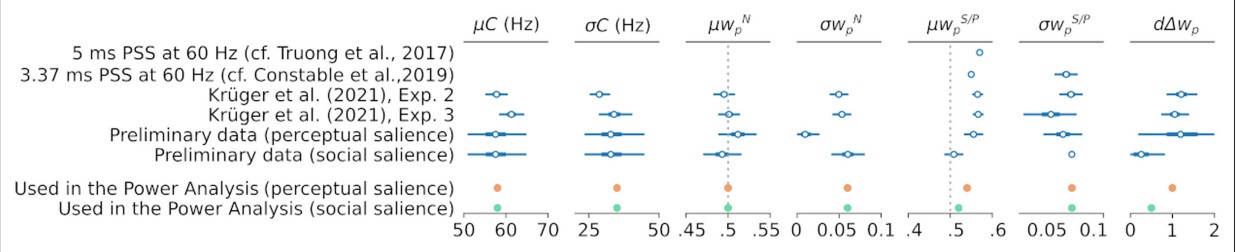

**Figure 10.** Parameter estimates and their respective uncertainties for social and perceptual salience.

'Bayesian power analysis' based on simulations as proposed by *Kruschke, 2014* and described for the TOJ-TVA model in Tünnermann et al. (2017; see also *Tünnermann, 2021*).

In this framework, power was estimated as posterior probability of obtaining a 'successful result'. This estimate is based on the proportion of observed successes in a large number of simulated experiments (here: 200). A success was defined by detecting the effect of interest, for instance, a certain change in parameter value following social association with a 95% highest density interval that excludes zero (no effect). Similar to a classic power analysis, these simulations required assumptions about effect sizes, which we fed into simulations as concrete parameter values (or differences between conditions) of interest. The simulations were further specified for specific sample and design configurations (e.g. number of participants, number of trials, SOA spacing, etc.) that determined the likelihood of obtaining powerful results, and for which power is estimated.

SOA spacing and number of trial repetitions were chosen based on previous studies using similar designs (*Krüger et al., 2021b*; *Krüger and Scharlau, 2021a*), and only the number of participants was varied to scale power. Model hyperpriors on $w_p$ ($\mu = 0.5$, $\sigma = 0.2$) and $C$ ($\mu = 0.08$, $\sigma = 0.05$) were informed by a number of previously published measurements (see *Tünnermann, 2021*, for details) but were kept vague enough to allow new data to drive changes in the posterior. Importantly, these priors represent neutral attentional weights; any shifts in the attentional weights by perceptual or social salience are driven by the data. The expected effects (and where possible, their corresponding uncertainties) were estimated based on findings from previously published results of (1) self-association (via ownership; Experiments 1 and 2 in *Truong et al., 2017* and Experiment 1 in *Constable et al., 2019*; the corresponding $w_p$ was determined by finding a value that approximately produces the reported

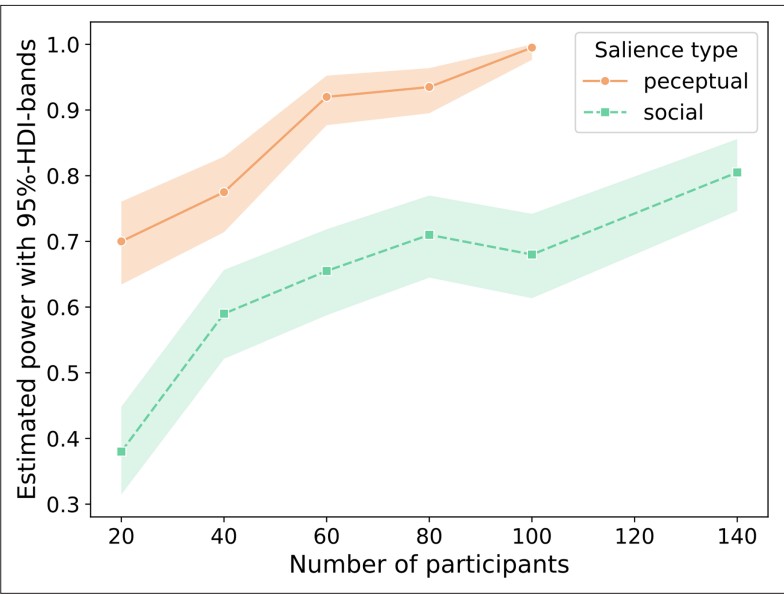

**Figure 11.** Estimated power for observing a reliable self-bias in attentional selection for the perceptual salience (orange solid line) and social salience (green dashed line) conditions. Shaded bands indicate their respective 95% highest density intervals.

PSS shifts, assuming a typical $C$ = 60 Hz), and (2) perceptual salience (Experiments 2 and 3 in *Krüger et al., 2021b*) on TOJs, as well as (3) estimates derived from 12 pilot participants to gauge effect sizes for both social and perceptual salience conditions (see *Figure 10*).

The pilot TOJ task included a perceptual salience, a social salience, and a baseline condition. The effect estimates selected for power analysis reflect a more conservative approximation of their expected effect size. In order to better understand how power increases with the number of participants, we tested how the chance of observing true social and perceptual biases in attentional selection, reflected in an increase in attentional weight in social and perceptual salience induction, changed across five different sample sizes (see *Figure 11*).

Simulation-based power estimation curves for social and perceptual salience conditions suggested that detecting a perceptual salience effect of the estimated size with a power of 0.92 [$HDI^{95}$: 0.88 to 0.95] would be reached with 60 participants. For the same number of participants, the power with which a social salience effect of the estimated size would be detected was 0.62 [$HDI^{95}$: 0.54 to 0.69], increasing to 0.71 [$HDI^{95}$: 0.65 to 0.77] with 80 participants. Note that power is specified for the case that the $HDI^{95}$ of $\Delta w$ did not include 0, while Bayesian parameter estimation will allow for the gradual quantification of the credibility of parameter values. In light of power estimation as well as resource limitations, we aimed to recruit 70 participants for each of the two experiments. Attentional selection effects of social salience were further quantified across both experiments, allowing the pooling of participants across experiments in order to address the primary research question with a total of 140 participants, affording a power of 0.81 [$HDI^{95}$: 0.75 to 0.86]. The other research questions were addressed with 69 and 71 participants each. Note that this is well in line with previous studies evidencing self-biases in early attentional selection via TOJs included between 36 and 78 participants per experiment (*Constable et al., 2019*: 48 participants in exp. 1; *Truong et al., 2017*: 73 and 78 participants in exp. 1 and 2, respectively; *Jublie and Kumar, 2021*: 36 and 33 participants in exp. 1 and 2, respectively).

## Study structure

The study consists of two experiments. Both followed a full within-participant design. Experiment 1 addressed research questions Q1 (social salience effect) and Q2 (processing level). Experiment 2 addressed research questions Q1 and Q3 (social vs perceptual salience). As both experiments were designed to assess the effects of social salience on attentional selection within the social decision dimension, data from both experiments was further pooled in a cross-experimental analysis in order to achieve higher power.

## Stimuli

Stimuli consisted of two arrays of 8 × 8 perceptual objects, one on the left hemifield of the screen and one on the right. Each array showed a different perceptual object. A perceptual object was defined as a shape (circle, square, triangle, hexagon) paired with a color (blue, green, red, yellow). The shapes that were paired with each other were matched in sizes (circles and squares: 1° vis. angle; triangles and hexagons: 1° vis. angle), and the colors that were paired with each other were matched in relative contrast ratio (blue and green: 1.93; red and yellow: 2.14) to the gray background (RGB 90, 90, 90). The 8 × 8 'background' shapes were placed on an even grid to avoid overlap and crowding, with small random offsets between trials to prevent adaptation. The four center locations of each array grid were possible locations where targets could appear. On each trial, two target shapes would appear, one in each hemifield, randomly at one of the four locations (counterbalanced). The distance to fixation was kept consistent, to prevent attentional biases due to target location. A target shape was the same perceptual shape that was present in the background shapes, but larger in size (1.3° vis. angle).

On all trials, except perceptual salience trials, the shapes were of the same color as the surrounding 'background' shapes. On perceptual salience trials, the color of one of the two target shapes was light gray, with an increased luminance contrast relative to the background of 4.79. This was the salient target and defined as the 'probe' on perceptual salience trials. The non-salient target was defined as the reference. On trials in which no perceptual salience was induced, the probe and reference were defined by their social association (probe = self; reference = other) or randomly as one of the two shapes (perceptual baseline). In the center of the screen, between the two arrays of perceptual

objects, was a central fixation cross, displayed in white, which participants were encouraged to fixate on during all trials. No additional eye tracking was implemented.

When responding, participants had the opportunity to move a mouse cursor to the selected perceptual object (perceptual decision dimension) or a social identity label (social decision dimension). In the first case, both shapes were shown in light gray above and below the fixation cross. In the second case, the labels 'self' and 'other' were presented above and below fixation. Which shape/ label was presented above and below fixation was randomly selected on each trial. The mouse cursor always started from fixation on each trial.

## Tasks

Each experiment consists of two tasks, which were both completed by all participants: TOJ and matching. The TOJ task was used in conjunction with the computational TVA to decompose attentional biases into model parameters of visual attention (*Tünnermann et al., 2015*; *Tünnermann et al., 2017*) and to answer the main research questions. The matching task provided individual measures of SPE magnitudes that are comparable with previous literature (e.g. *Sui et al., 2012*). SPE quantification from the matching task was used to confirm whether the included participants show self-prioritization with the current stimuli in a well-established paradigm, and to what degree early temporal self-biases allow to explain individual SPE magnitudes via correlations.

### TOJ task

On each trial, a probe and a reference stimulus were shown on the left and right sides of a multi-array shape display (8×8 array of shapes). The 'target shapes' were 30% bigger than the 'background shapes' and were shown in one of four possible locations that were symmetrical to the fixation cross. The target and background shapes were shown for an initial 200 ms to allow attentional capacity to build up across the visual field (*Krüger and Scharlau, 2021a*). One shape was shown on one side, while the other shape was shown on the opposite side. Lateralization of the shapes was randomly selected on each trial. After the initial 200 ms, the target shapes were flickered simultaneously or with one of 10 different levels of stimulus onset asynchronies (SOAs: [−83,−63, −42,−28, −14, 14, 28, 42, 63, 83] ms). Afterwards, participants had to indicate either which shape flickered first, by selecting one of the two shapes that were shown above and below a fixation cross (perceptual decision dimension), or one of two identity labels that corresponded to the associated shapes (social decision dimension). The location of the response shapes and response identity labels was chosen to be non-lateralized and randomly altered between trials to reduce any possible location bias effects. Each SOA was presented multiple times, between 8 and 24 times (see *Figure 3*). Higher repetitions at smaller SOAs allowed an enhanced precision in an area of the psychometric function that was highly informative for TVA.

### Matching task

The matching task was conducted between two parts of the TOJ task in order to create associations between shapes and social identities, and to practice and consolidate these associations. At the beginning of this task, participants were told that one of the two geometric shapes that was used in the TOJ task has been assigned to them, and the other shape has been assigned to another participant in the experiment – someone they did not know, but who was of similar age and gender. The shapes that were used for this task were colored gray to avoid associating identities with colors rather than with shape. During the task, they were shown one of the shapes on screen, paired with a label ('you' or 'other'). Their task was to report whether the shape-label pair matched or mismatched in their associated identity. The shape would either appear above or below the fixation cross, and the label would always appear in the opposite location. The location of presentation was counterbalanced across trials. Overall, the task followed a 2 × 2 design: shape-label pairing (match/mismatch) and social identity (self/other). Only matching trials were analyzed to derive the SPE.

## Conditions

In Experiment 1, the TOJ task was conducted in three different conditions:

- A **baseline condition**, which was conducted prior to the social association of the stimuli and required participants to judge which shape flickered first (perceptual decision dimension).

- A **social salience condition** in which participants had to judge temporal order based on the shape-associated identity (social salience, social decision dimension).
- A **social salience condition** in which participants had to judge temporal order based on the shape (social salience, perceptual decision dimension).

For the analysis, the probe across all conditions was defined as the shape that was associated with the self, both before and after association. Note that the participant was not aware of this, and it did not affect the stimulus presentation in any way. However, it changes the interpretation of effect directionalities. Here, the probe is always a specific shape and can indicate shape biases (such as shape preferences) in the baseline condition and social biases in the social salience condition. As such, the baseline condition allowed to control for any individual-specific pre-existing biases.

In Experiment 2, the TOJ task was conducted in four different conditions:

- A **baseline condition**, which was conducted prior to the social association of the stimuli and required participants to judge which shape flickered first (perceptual decision dimension).
- A **perceptual salience condition**, which was conducted prior to the social association of the stimuli and induced local salience by altering the color of the target relative to the background. Participants had to judge the temporal order of the flicker based on the shape (perceptual salience, perceptual decision dimension).
- A **social salience condition** in which participants had to judge temporal order based on the shape-associated identity (social salience, social decision dimension).
- A **social + perceptual salience condition** in which participants had to judge temporal order based on the shape-associated identity (social salience, social decision dimension), while local perceptual salience was induced by altering the color of either the self- or other-associated stimulus relative to the background pattern. This condition was split into two different forms: those where perceptual salience was present on self-associated shapes, and those where perceptual salience was present on other-associated shapes.

For the analysis, the probe in the social salience condition was defined as the self-associated shape. Similarly, the baseline condition was recorded to provide a suitable baseline for the social salience condition, defining the self-associated shape as a probe. In the other baseline condition, as well as the perceptual salience and social + perceptual salience conditions, the probe was on approximately 50% of trials defined as the self-shape, and on the other trials as the other-shape. This was done as both the self- and other-associated shapes had an equal probability of being perceptually salient. This was important, as not to bias social relevance by perceptual salience, setting up wrong predictions about an enhanced perceptual salience of socially salient shapes. In the perceptual salience and social + perceptual salience conditions, the probe was always the salient shape.

As social associations were induced halfway through the experiment, the baseline condition was always conducted first. In Experiment 2, the perceptual salience condition was also conducted prior to social association. This order allowed us to retain the same perceptual objects in all conditions. To consolidate social association learning and to provide an individual measure of self-prioritization that is comparable with the literature, all participants completed a perceptual matching task after the baseline condition. In Experiment 1, the order of the two social salience conditions was blocked and counterbalanced across participants. In Experiment 2, the social salience and social + perceptual salience conditions were intermixed. Self-associated salient and other-associated salient trials were intermixed as well.

In each experiment, all participants completed all conditions. To enhance direct comparability, all conditions were conducted on the same day within the same session. Participants were not made aware of the hypotheses of the study.

## Counterbalancing

The associations between social identities and specific shapes were counterbalanced across participants. In the matching task, response keys that indicate a 'match' or 'mismatch' were also counterbalanced across participants. In the TOJ task, the condition order of the different decisional dimensions of the social salience condition was counterbalanced. Within the TOJ task, the following parameters were randomized (r) or counterbalanced (c) within and across blocks:

- Target hemifield: Whether the probe is shown left or right. (c)

- Stimulus location: Which out of four possible locations in each hemifield the probe is presented. Probe and reference are symmetrically aligned to keep the distance to fixation equal for each trial. (c)
- Response label position: Whether the probe label is presented above or below fixation. Probe and reference labels are always presented opposite each other vertically. (c)
- Order of SOAs: Whether, and by which amount, the probe precedes or follows the presentation of the reference. (r)
- Choice of probe: Which stimulus is defined as the probe in perceptual salience and perceptual baseline condition on each trial (r). The probe is determined by the shape for the social salience and social baseline conditions.

Within the matching task, the following parameters were randomized (r) or counterbalanced (c) within and across blocks:

- Condition order: Which stimulus pair (identity, matching) is shown next (r)
- Stimulus location: Whether the shape is shown above or below fixation. The label is always shown in the vertically opposite position (c)
- Stimulus frequency: Whether each identity and matching condition is shown with the same frequency within and across blocks (c)

### Indices, parameter estimation, and analysis

The main outcome of the TOJ task was the proportion of responses with which the probe was judged to flicker first. Based on the TVA-TOJ framework, the following parameters were estimated:

- **Processing capacity** $C$: refers to the overall speed of processing in Hz and is estimated for each participant separately and is typically found to be at 60 Hz on average with stimuli similar to the ones of the planned study.
- **Attentional weight** $w_p$: describes the weight that is given to the probe stimulus during attentional selection and is estimated for each participant and condition separately baseline: $w_p^N$ ; salience: ($w_p^{S/P}$). Attentional weights of 0.5 indicate that equal weight is given to either stimulus. Increases in this attentional weight indicate biases toward the salient (probe) stimulus. Baseline-corrected shifts in attentional weights as a result of salience induction are reflected in $w_{p,effect}^{S/P}$.
- **Encoding rates** $v_p$ **and** $v_r$: indicate the speeds with which each of the targets (probe and reference, respectively) are encoded into visual short-term memory.

These parameters were estimated by embedding the TVA-based TOJ equation for the probability of reporting the probe stimulus as appearing first ($P_p^{1st}$, see *Equation 1*; cf. *Tünnermann et al., 2015*) in hierarchical Bayesian models (detailed in *Figures 4–6*):

$$P_{p1st}\left(v_p, v_r, \text{SOA}\right) = \begin{cases} 1 - e^{-v_p|SOA|} + e^{v_p|SOA|}\dfrac{v_p}{v_p + v_r} & \text{if SOA} < 0 \\[2ex] e^{-v_r|SOA|}\dfrac{v_p}{v_p + v_r} & \text{otherwise ,} \end{cases} \tag{1}$$

where $v_p$ and $v_r$ represent the processing rates of probe and reference, SOA is the interval between probe and reference (negative if probe leads). The models were estimated with PyMC (*Abril-Pla et al., 2023*) using the NUTS sampler (*Hoffman and Gelman, 2014*). 30,000 posterior samples were drawn after 1000 tuning samples. Further details can be found at: https://osf.io/ehu75. The main outcome of the matching task was the accuracy toward matching shapes and labels. SPE magnitude was extracted as the differential accuracy between self-associated and other-associated information.

## Acknowledgements

The study was supported by grants from the Leverhulme Foundation (REF RPG-2019-010) and the Institutional Research Leave Award (CF10831-27) to JS.

## Additional information

### Funding

| Funder | Grant reference number | Author |
|---|---|---|
| Leverhulme Trust | REF RPG-2019-010 | Jie Sui |
| University of Aberdeen | CF10831-27 | Jie Sui |

The funders had no role in study design, data collection and interpretation, or the decision to submit the work for publication.

### Author contributions

Meike Scheller, Conceptualization, Resources, Data curation, Software, Formal analysis, Supervision, Validation, Investigation, Visualization, Methodology, Writing – original draft, Project administration, Writing – review and editing; Jan Tünnermann, Conceptualization, Resources, Software, Formal analysis, Validation, Investigation, Visualization, Methodology, Writing – review and editing; Katja Fredriksson, Huilin Fang, Data curation, Investigation, Writing – review and editing; Jie Sui, Conceptualization, Resources, Supervision, Funding acquisition, Writing – review and editing

### Author ORCIDs

Meike Scheller ⬤ https://orcid.org/0000-0002-3021-5614

### Ethics

The study received ethical approval from the University of Aberdeen Psychology ethics committee (PEC/4541/2020/9). All participants provided written, informed consent prior to taking part in the study.

Reviewer #1 (Public review): https://doi.org/10.7554/eLife.100932.3.sa1
Reviewer #2 (Public review): https://doi.org/10.7554/eLife.100932.3.sa2
Author response https://doi.org/10.7554/eLife.100932.3.sa3

## Additional files

### Supplementary files

MDAR checklist

### Data availability

Research questions, analyses, models, and additional details have been preregistered on the Open Science Framework: https://osf.io/ehu75. Analysis notebooks and data are also available on the OSF project repository https://osf.io/a62df.

The following previously published datasets were used:

| Author(s) | Year | Dataset title | Dataset URL | Database and Identifier |
|---|---|---|---|---|
| Scheller M, Tünnermann J, Fredriksson K, Sui J | 2021 | Self-relevance and perceptual salience in early attentional processing | https://osf.io/ehu75 | Open Science Framework, ehu75 |
| Scheller M, Tünnermann J, Fredriksson K, Sui J | 2026 | Self-relevance and perceptual salience in early attentional processing | https://osf.io/a62df/ | Open Science Framework, a62df |

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
