## [Editor Report · eLife Assessment]

This study presents a **valuable** finding on the mechanism of self-prioritization by revealing the influence of self-associations on early attentional selection. The evidence supporting the claims of the authors is **solid**, although inclusion of a discussion about the generalization and limitation would have strengthened the study. The work will be of interest to researchers in psychology, cognitive science, and neuroscience.

---

## [Referee Report · Reviewer #1 (Public review)]

Summary:

The authors intended to investigate the earliest mechanisms enabling self-prioritization, especially in the attention. Combining a temporal order judgement task with computational modelling based on the Theory of Visual Attention (TVA), the authors suggested that the shapes associated with the self can fundamentally alter the attentional selection of sensory information into awareness. This self-prioritization in attentional selection occurs automatically at early perceptual stages. Furthermore, the processing benefits obtained from attentional selection via self-relatedness and physical salience were separated from each other.

Strengths:

The manuscript is written in a way that is easy to follow. The methods of the paper are very clear and appropriate.

Comments on revisions:

The authors clearly showed the relationship between attention and self-prioritization.

---

## [Referee Report · Reviewer #2 (Public review)]

Summary:

The main aim of this research was to explore whether and how self-associations (as opposed to other-associations) bias early attentional selection, and whether this can explain well-known self-prioritization phenomena, such as the self-advantage in perceptual matching tasks. The authors adopted the Visual Attention Theory (VAT) by estimating VAT parameters using a hierarchical Bayesian model from the field of attention and applied it to investigate the mechanisms underlying self-prioritization. They also discussed the constraints on the self-prioritization effect in attentional selection. The key conclusions reported were: (1) self-association enhances both attentional weights and processing capacity, (2) self-prioritization in attentional selection occurs automatically but diminishes when active social decoding is required, and (3) social and perceptual salience capture attention through distinct mechanisms.

Strengths:

Transferring the Theory of Visual Attention parameters estimated by a hierarchical Bayesian model to investigate self-prioritization in attentional selection was a smart approach. This method provides a valuable tool for accessing the very early stages of self-processing, i.e., the attention selection. The authors conclude that self-associations can bias visual attention by enhancing both attentional weights and processing capacity, and that this process occurs automatically. These findings offer new insights into the self-prioritization from the perspective of early stage of attentional selection.

Weaknesses:

The results are still not convincing enough to definitively support their conclusions. The generalization of the findings needs further examination. Whether this attentional selection mechanism of self-prioritization can be generalized to other stimuli, such as self-name, self-face, or other domains of self-association advantages, remains to be tested. More empirical data are needed.

---

## [Author Response]

The following is the authors’ response to the original reviews.

**Public Reviews:**

**Reviewer #1 (Public review):**
Summary:The authors intended to investigate the earliest mechanisms enabling self-prioritization, especially in the attention. Combining a temporal order judgement task with computational modelling based on the Theory of Visual Attention (TVA), the authors suggested that the shapes associated with the self can fundamentally alter the attentional selection of sensory information into awareness. This self-prioritization in attentional selection occurs automatically at early perceptual stages. Furthermore, the processing benefits obtained from attentional selection via self-relatedness and physical salience were separated from each other.Strengths:The manuscript is written in a way that is easy to follow. The methods of the paper are very clear and appropriate.

Thank you for your valuable feedback and helpful suggestions. Please see specific answers below.

Weaknesses:There are two main concerns:(1) The authors had a too strong pre-hypothesis that self-prioritization was associated with attention. They used the prior entry to consciousness (awareness) as an index of attention, which is not appropriate. There may be other processing that makes the stimulus prior to entry to consciousness (e.g. high arousal, high sensitivity), but not attention. The self-related/associated stimulus may be involved in such processing but not attention to make the stimulus easily caught. Perhaps the authors could include other methods such as EEG or MEG to answer this question.

We found the possibility of other mechanisms to be responsible for “prior entry” interesting too, but believe there are solid grounds for the hypothesis that it is indicative of attention:

First, prior entry has a long-standing history as in index of attention (e.g., Titchener, 1903; Shore et al., 2001; Yates and Nicholls, 2009; Olivers et al. 2011; see Spence & Parise, 2010, for a review.) Of course, other factors (like the ones mentioned) can contribute to encoding speed. However, for the perceptual condition, we systematically varied a stimulus feature that is associated with selective attention (salience, see e.g. Wolfe, 2021) and kept other features that are known to be associated with other factors such as arousal and sensitivity constant across the two variants (e.g. clear over threshold visibility) or varied them between participants (e.g. the colours / shapes used).

Second, in the social salience condition we used a manipulation that has repeatedly been used to establish social salience effects in other paradigms (e.g., Li et al., 2022; Liu & Sui, 2016; Scheller et al., 2024; Sui et al., 2015; see Humphreys & Sui, 2016, for a review). We assume that the reviewer’s comment suggests that changes in arousal or sensitivity may be responsible for social salience effects, specifically. We have several reasons to interpret the social salience effects as an alteration in attentional selection, rather than a result of arousal or sensitivity:

Arousal and attention are closely linked. However, within the present model, arousal is more likely linked to the availability of processing resources (capacity parameter *C*). That is, enhanced arousal is typically not stimulus-specific, and therefore unlikely affects the *relative* advantage in processing weights/rates of the self-associated (vs other-associated) stimuli. Indeed, a recent study showed that arousal does not modulate the relative division of attentional resources (as modelled by the Theory of Visual Attention; Asgeirsson & Nieuwenhuis, 2017). As such, it is unlikely that arousal can explain the observed results in relative processing changes for the self and other identities.

Further, there is little reason to assume that presenting a different shape enhances perceptual sensitivity. Firstly, all stimuli were presented well above threshold, which would shrink any effects that were resulting from increases in sensitivity alone. Secondly, shape-associations were counterbalanced across participants, reducing the possibility that specific features, present in the stimulus display, lead to the measurable change in processing rates as a result of enhanced shape-sensitivity.

Taken together, both, the wealth of literature that suggests prior entry to index attention and the specific design choices within our study, strongly support the notion that the observed changes in processing rates are indicative of changes in attentional selection, rather than other mechanisms (e.g. arousal, sensitivity).

(2) The authors suggested that there are two independent attention processes. I suspect that the brain needs two attention systems. Is there a probability that the social and perceptual (physical properties of the stimulus) salience fired the same attention processing through different processing?

We appreciate this thought-provoking comment. We conceptualize attention as a process that can facilitate different levels of representation, rather than as separate systems tuned to specific types of information. Different forms of representation, such as the perceptual shape, or the associated social identity, may be impacted by the same attentional process at different levels of representation. Indeed, our findings suggest that both social and perceptual salience effects may result from the same attentional system, albeit at different levels of representation. This is further supported by the additivity of perceptual and social salience effects and the negative correlation of processing facilitations between perceptually and socially salient cues. These results may reflect a trade-off in how attentional resources are distributed between either perceptually or socially salient stimuli.

**Reviewer #2 (Public review):**
Summary:The main aim of this research was to explore whether and how self-associations (as opposed to other associations) bias early attentional selection, and whether this can explain well-known self-prioritization phenomena, such as the self-advantage in perceptual matching tasks. The authors adopted the Visual Attention Theory (VAT) by estimating VAT parameters using a hierarchical Bayesian model from the field of attention and applied it to investigate the mechanisms underlying self-prioritization. They also discussed the constraints on the self-prioritization effect in attentional selection. The key conclusions reported were:(1) Self-association enhances both attentional weights and processing capacity(2) Self-prioritization in attentional selection occurs automatically but diminishes when active social decoding is required, and(3) Social and perceptual salience capture attention through distinct mechanisms.Strengths:Transferring the Theory of Visual Attention parameters estimated by a hierarchical Bayesian model to investigate self-prioritization in attentional selection was a smart approach. This method provides a valuable tool for accessing the very early stages of self-processing, i.e., attention selection. The authors conclude that self-associations can bias visual attention by enhancing both attentional weights and processing capacity and that this process occurs automatically. These findings offer new insights into self-prioritization from the perspective of the early stage of attentional selection.

Thank you for your valuable feedback and helpful suggestions. Please see specific answers below.

Weaknesses:(1) The results are not convincing enough to definitively support their conclusions. This is due to inconsistent findings (e.g., the model selection suggested condition-specific c parameters, but the increase in processing capacity was only slight; the correlations between attentional selection bias and SPE were inconsistent across experiments), unexpected results (e.g., when examining the impact of social association on processing rates, the other-associated stimuli were processed faster after social association, while the self-associated stimuli were processed more slowly), and weak correlations between attentional bias and behavioral SPE, which were reported without any p-value corrections. Additionally, the reasons why the attentional bias of self-association occurs automatically but disappears during active social decoding remain difficult to explain. It is also possible that the self-association with shapes was not strong enough to demonstrate attention bias, rather than the automatic processes as the authors suggest. Although these inconsistencies and unexpected results were discussed, all were post hoc explanations. To convince readers, empirical evidence is needed to support these unexpected findings.

Thank you for outlining the specific points that raise your concern. We were happy to address these points as follows:

a. Replications and Consistency: In our study, we consistently observed trends (relative reduction in processing speed of the self-associated stimulus) in the social salience conditions across experiments. While Experiment 2 demonstrated a significant reduction in processing rate towards self-stimuli, there was a notable trend in Experiment 1 as well.

b. Condition-specific parameters**:** The condition-specific C parameters, though presenting a small effect size, significantly improved model fit. Inspecting the HDI ranges of our estimated C parameters indicates a high probability (85-89%) that processing capacity increased due to social associations, suggesting that even small changes (~2Hz) can hold meaningful implications within the context attentional selection.

Please also note that the main conclusions about relative salience (self/other, salient/non-salient) are based on the relative processing rates. Processing rates are the product of the processing capacity (condition- but not stimulus dependent) and the attentional weight (condition and stimulus dependent). The latter is crucial to judge the *relative* advantage of the salient stimulus. Hence, the self-/salient stimulus advantage that is reflected in the ‘processing rate difference’ is automatically also reflected in the relative attentional weights attributed to the self/other and salient/non-salient stimuli. As such, the overall results of an automatic relative advantage of self-associated stimuli hold, independently of the change in overall processing capacity.

c. Correlations: Regarding the correlations the reviewer noted, we wish to clarify that these were exploratory, and not the primary focus of our research. The aim of these exploratory analyses was to gauge the contribution of attentional selection to matching-based SPEs. As SPEs measured via the matching task are typically based on multiple different levels of processing, the contribution of early attentional selection to their overall magnitude was unclear. Without being able to gauge the possible effect sizes, corrected analyses may prevent detecting small but meaningful effects. As such, the effect sizes reported serve future studies to estimate power a priori and conduct well-powered replications of such exploratory effects. Additionally, Bayes factors were provided to give an appreciation of the strength of the evidence, all suggesting at least moderate evidence in favour of a correlation. Lastly, please note that effects that were measured within individuals and task (processing rate increase in social and perceptual decision dimensions in the TOJ task) showed consistent patterns, suggesting that the modulations within tasks were highly predictive of each other, while the modulations between tasks were not as clearly linked. We will add this clarification to the revised manuscript.

d. Unexpected results: The unexpected results concerning the processing rates of other-associated versus self-associated stimuli certainly warrant further discussion. We believe that the additional processing steps required for social judgments, reflected in enhanced reaction times, may explain the slower processing of self-associated stimuli in that dimension. We agree that not all findings will align with initial hypotheses, and this variability presents avenues for further research. We have added this to the discussion of social salience effects.

e. Whether association strength can account for the findings: We appreciate the scepticism regarding the strength of self-association with shapes. However, our within-participant design and control matching task indicate that the relative processing advantage for self-associated stimuli holds across conditions. This makes the scenario that “the self-association with shapes was not strong enough to demonstrate attention bias” very unlikely. Firstly, the relative processing advantage of self-associated stimuli in the perceptual decision condition, and the absence of such advantage in the social decision condition, were evidenced in the same participants. Hence, the strength of association between shapes and social identities was the same for both conditions. However, we only find an advantage for the self-associated shape when participants make perceptual (shape) judgements. It is therefore highly unlikely that the “association strength” can account for the difference in the outcomes between the conditions in experiment 1. Also, note that the order in which these conditions were presented was counter-balanced across participants, reducing the possibility that the automatic self-advantage was merely a result of learning or fatigue. Secondly, all participants completed the standard matching task to ascertain that the association between shapes and identities did indeed lead to processing advantages (across different levels).

In summary, we believe that the evidence we provide supports the final conclusions. We do, of course, welcome any further empirical evidence that could enhance our understanding of the contribution of different processing levels to the SPE and are committed to exploring these areas in future work.

(2) The generalization of the findings needs further examination. The current results seem to rely heavily on the perceptual matching task. Whether this attentional selection mechanism of self-prioritization can be generalized to other stimuli, such as self-name, self-face, or other domains of self-association advantages, remains to be tested. In other words, more converging evidence is needed.

The reviewer indicates that the current findings heavily rely on the perceptual matching task, and it would be more convincing to include other paradigm(s) and different types of stimuli. We are happy to address these points here: first, we specifically used a temporal order paradigm to tap into specific processes, rather than merely relying on the matching task. Attentional selection is, along with other processes, involved in matching, but the TOJ-TVA approach allows tapping into attentional selection specifically. Second, self-prioritization effects have been replicated across a wide range of stimuli (e.g. faces: Wozniak et al., 2018; names or owned objects: Scheller & Sui, 2022a, or even fully unfamiliar stimuli: Wozniak & Knoblich, 2019) and paradigms (e.g. matching task: Sui et al., 2012; cross-modal cue integration: e.g. Scheller & Sui, 2022b; Scheller et al., 2023; continuous flash suppression: Macrae et al., 2017; temporal order judgment: Constable et al., 2019; Truong et al., 2017). Using neutral geometric shapes, rather than faces and names, addresses a key challenge in self research: mitigating the influence of stimulus familiarity on results. In addition, these newly learned, simple stimuli can be combined with other paradigms, such as the TOJ paradigm in the current study, to investigate the broader impact of self-processing on perception and cognition.

To the best of our knowledge, this is the first study showing evidence about the mechanisms that are involved in early attentional selection of socially salient stimuli. Future replications and extensions would certainly be useful, as with any experimental paradigm.

(3) The comparison between the "social" and "perceptual" tasks remains debatable, as it is challenging to equate the levels of social salience and perceptual salience. In addition, these two tasks differ not only in terms of social decoding processes but also in other aspects such as task difficulty. Whether the observed differences between the tasks can definitively suggest the specificity of social decoding, as the authors claim, needs further confirmation.

Equating the levels of social and perceptual salience is indeed challenging, but not an aim of the present study. Instead, the present study directly compares the mechanisms and effects of social and perceptual salience, specifically experiment 2. By manipulating perceptual salience (relative colour) and social salience (relative shape association) independently and jointly, and quantifying the effects on processing rates, our study allows to directly delineate the contributions of each of these types of salience. The results suggest additive effects (see also Figure 7). Indeed, the possibility remains that these effects are additive because of the use of different perceptual features, so it would be helpful for future studies to explore whether similar perceptual features lead to (supra-/sub-) additive effects. In either case, the study design allows to directly compare the effects and mechanisms of social and perceptual salience.

Regarding the social and perceptual decision dimensions, they were not expected to be equated. Indeed, the social decision dimension requires additional retrieval of the associated identity, making it likely more challenging. This additional retrieval is also likely responsible for the slower responses towards the social association compared to the shape itself. However, the motivation to compare the effects of these two decisional dimensions lies in the assumption that the self needs to be task relevant. Some evidence suggests that the self needs to be task-relevant to induce self-prioritization effects (e.g., Woźniak & Knoblich, 2022). However, these studies typically used matching tasks and were powered to detect large effects only (e.g. *f* = 0.4, *n* = 18). As it is likely that lacking contribution of decisional processing levels (which interact with task-relevance) will reduce the SPE, smaller self-prioritization effects that result from earlier processing levels may not be detected with sufficient statistical power. Targeting specific processing levels, especially those with relatively early contributions or small effect sizes, requires larger samples (here: *n* = 70) to provide sufficient power. Indeed, by contrasting the relative attentional selection effects in the present study we find that the self does not need to be task-relevant to produce self-prioritization effects. This is in line with recent findings of prior entry of self-faces (Jubile & Kumar, 2021)

**Reviewer #2 (Recommendations for the authors):**
Suggestions:(1) The research questions should be revised to better align with the conclusions. For example, Q2 is phrased as "Does self-relatedness bias attentional selection at the level of the perceptual feature representation (shape) or at the level of the associated identity (social association)," which is unclear in its reference to "levels." A more appropriate phrasing would be whether the self-association bias occurs automatically or whether it depends on explicit social decoding.

Thank you for this suggestion – we have revised the phrasing accordingly: “Does self-relatedness bias attentional selection automatically or does it require explicit social decoding?”

(2) After presenting the data, it would be helpful to include one or two sentences summarizing the conclusions drawn from the data and how they relate to the research questions. Currently, readers are left to guess whether the results are consistent with the hypotheses.

Thank you for this suggestion, which we think will enhance the clarity of the manuscript – we have added summary sentences when presenting the results:

“This cross-experimental parameter inspection revealed that participants exhibited an attentional selection bias towards socially associated information. Interestingly, enhanced processing speed was observed for other-associated rather than self-associated information, a pattern that diverged from our prediction.”

(1) “Results from experiment 2 demonstrated a faster, more automatic attentional selection for self-associated information when the decision did not require explicit social decoding. When the social identity had to be judged, processing speed for self-associated information decreased. Contrary to the hypothesis that social decoding is necessary for self-prioritization to emerge, these findings suggest that attentional selection can operate automatically to prioritize self-associated information. “

(2) “Taken together, as also confirmed in the cross-experimental analysis, attentional selection favoured the other-related information when social identity had to be judged. In contrast, perceptual salience, as predicted, led to increased processing speed for the more salient stimulus. “

(3) The identity of the "other" used in the experiments is unclear, making it uncertain whether the results are self-specific. It would be beneficial to compare the self condition with a control condition, such as a close friend vs. an unfamiliar other. Alternatively, the results may reflect attentional bias for familiar vs. unfamiliar individuals rather than self-specific bias.

Thank you for this comment. Firstly, we would like to clarify that we have provided participants with a description of who the “other” is (see methods: “At the beginning of this task, participants were told that one of the two geometric shapes that was used in the TOJ task has been assigned to them, and the other shape has been assigned to another participant in the experiment – someone they did not know, but who was of similar age and gender”). We aimed to make the ‘other’ as concrete as possible, while maintaining a ‘stranger’ identity.

Secondly, this specification is in line with the vast majority of the literature, which typically measures the effects of self-prioritization relative to the association with an unfamiliar other (stranger), or an unfamiliar and familiar other (e.g. friend, family member). They find that processing advantages that affect friend-related stimuli (friend-stimuli being processed faster than stranger-associated stimuli) are likely mediated by self-extension, that is, an association of the friend with the self. As such, SPEs, relative to familiar others, are typically smaller in size (see, e.g., Sui et al., 2012). They, however, are less stable and more variable than the self-prioritization effects measured relative to a stranger (see Scheller & Sui, 2022 JEP:HPP). Importantly, this is driven by the variability of the friend-associated stimulus, rather than the self or other-associated stimulus (see Figure 4 in main text and S5 in supplementary material in Scheller & Sui, 2022: here). Effectively, this would suggest that choosing a familiar other as a reference would not only (a) lead to a smaller effect size, but also (b) be a less stable effect, which likely depends on the association the individual has to the other familiar person. In contrast, by associating the other shape with another participant in this experiment, we provide participants not only with a concrete representation of a stranger, but also maximise our ability to detect true effects, as these are likely to be larger and more stable.

(4) The key aspects of the procedure (e.g., the order of different conditions) and its rationale need to be clearly explained before or during the presentation of the results. Currently, readers are left to infer certain details.

Thank you for pointing this out. The methods that provide these details are outlined at the end of the document, however, we agree it would be useful to bring some of these details up. We have therefore revised the methods figure (Figure 3) to include an outline of the task type, order, and trial numbers. Task boxes are colour coded by the conditions that are listed in the results figures of the manuscript. We also added these details to the caption of Figure 3.

“Task structures of Experiments 1 and 2. Both experiments started with a TOJ baseline task. In Experiment 1, only non-salient targets were presented, while in Experiment 2, perceptually salient and non-salient trials were included. These were presented in randomly intermixed order. Next, targets were associated with social identities. Associations were practiced using the matching task. Following association learning, which attaches social salience to the shapes, participants completed the same TOJ task as before. In Experiment 1, they completed one block using a social decision dimension, and one block using a perceptual decision dimension. The order of these blocks was counterbalanced across participants to reduce the influence of order effects in the results. In Experiment 2, perceptually salient and non-salient stimuli were presented in an intermixed fashion, and participants responded within the social decision dimension. Each task block was preceded by 8 (matching) to 14 (TOJ) practice trials.”

(5) Certain imprecise terms used to describe the results, such as "slightly," "roughly," and "loosely," create confusion for the readers. The authors should take a clearer stance on the results and provide an explanation for why the data only "slightly," "roughly," or "loosely" support the findings.

Thank you for highlighting this. We have provided a more concrete wording and details throughout (e.g., “target shapes’ were 30% bigger than the ‘background shapes”).

Lastly, we have updated the formatting of the manuscript to provide higher fidelity figures, which were previously compromised by file conversion.